# DeepSplit: Scalable Verification of Deep Neural Networks via Operator Splitting

## Abstract

Analyzing the worst-case performance of deep neural networks against input perturbations amounts to solving a large-scale non-convex optimization problem, for which several past works have proposed convex relaxations as a promising alternative. However, even for reasonably-sized neural networks, these relaxations are not tractable, and so must be replaced by even weaker relaxations in practice. In this work, we propose a novel operator splitting method that can directly solve a convex relaxation of the problem to high accuracy, by splitting it into smaller sub-problems that often have analytical solutions. The method is modular, scales to very large problem instances, and compromises of operations that are amenable to fast parallelization with GPU acceleration. We demonstrate our method in obtaining tighter bounds on the worst-case performance of large convolutional networks in image classification and reinforcement learning settings.

## 1 Introduction

Despite their superior performance, neural networks lack formal guarantees, raising serious concerns about their adoption in safety-critical applications such as autonomous vehicles (Cao et al., 2019) and medical machine learning (Finlayson et al., 2019). Motivated by this drawback, there has been an increasing interest in developing tools to verify desirable properties for neural networks, such as robustness to adversarial attacks. Neural network verification refers to the problem of verifying whether the output of a neural network satisfies certain properties for a bounded set of input perturbations. This problem can be framed as optimization problems of the form

$$\text{minimize} \quad J(f(x)) \quad \text{subject to} \quad x \in \mathcal{X}, \tag{1}$$

where $f$ is given by a deep neural network, $J$ is a real-valued function representing a performance measure (or a specification), and $\mathcal{X}$ is a set of inputs to be verified. In this formulation, verifying the neural network amounts to certifying whether the optimal value of (1) is bounded below by a certain threshold. This problem is large-scale and non-convex, making it extremely difficult to solve efficiently–both in terms of time and memory. For ReLU activation functions and linear objectives, the problem in (1) can be cast as a Mixed-Integer Linear Program (MILP) (Lomuscio & Maganti, 2017; Cheng et al., 2017; Dutta et al., 2018; Fischetti & Jo, 2018), which can be solved for the global solution via, for example, Branch-and-Bound (BaB) methods. While we do not expect these approaches to scale to large problems, for small neural networks they can still be practical.

Instead of solving (1) for its global minimum, one can instead find *guaranteed lower bounds* on the optimal value via convex relaxations, such as Linear Programming (LP) (Wong & Kolter, 2017) and Semidefinite Programming (SDP) (Raghunathan et al., 2018; Fazlyab et al., 2019; 2020). Verification methods based on convex relaxations are sound but incomplete, i.e., they are guaranteed to detect all false negatives but also produce false positives, whose rate depends on the tightness of the relaxation. Although convex relaxations are polynomial-time solvable (in terms of number of decision variables), in practice they are not computationally tractable for large-scale neural networks. To improve scalability, these relaxations must typically be further relaxed (Wong & Kolter, 2017; Weng et al., 2018; Zhang et al., 2018; Dvijotham et al., 2018; Bunel et al., 2020a).

**Contributions** In this work, we propose an algorithm for solving convex relaxations *exactly* for large-scale neural networks. Our starting point is to express (1) as a constrained optimization problem whose constraints are imposed by the forward passes in the network. We then introduce additional decision variables and consensus constraints that naturally split the corresponding problem into independent subproblems, which often have closed-form solutions. Finally, we employ an operator splitting technique based on the Alternating Direction Method of Multipliers (ADMM) (Boyd et al., 2011), to solve the corresponding Lagrangian relaxation of the problem. This approach has several favorable properties. First, the method requires minimal parameter tuning and relies on simple operations, which scale to very large problems and can achieve a good trade-off between runtime and solution accuracy. Second, all the solver operations are amenable to fast parallelization with GPU acceleration. Third, our method is fully modular and applies to standard network architectures.

We employ our method to compute exact solutions to LP relaxations on the worst-case performance of adversarially trained deep networks, with a focus on networks whose convex relaxations were previously impossible to solve exactly due to their size. Specifically, we perform extensive experiments in the $\ell_\infty$ perturbation setting, where we verify robustness properties of image classifiers for CIFAR10 and deep Q-networks (DQNs) in Atari games (Zhang et al., 2020). Our method is able to quickly solve LP relaxations at scales that are too large for exact BaB verifiers, SDP relaxations, or commercial LP solvers such as Gurobi.

## 1.1 Related Work

**Convex relaxations** LP relaxations are relatively the most scalable form of convex relaxations (Ehlers, 2017). However, even solving LPs can become computationally prohibitive for small convolutional networks (Salman et al., 2019). One line of work studies computationally cheaper but looser bounds of the LP relaxation (Wong & Kolter, 2017), which have been extended to larger and more general networks and settings (Wong et al., 2018; Weng et al., 2018; Zhang et al., 2018; Xu et al., 2020). These bounds tend to be loose unless optimized during training, which typically comes at a significant cost to standard performance. Further work has aimed to tighten these bounds (Singh et al., 2019; Tjandraatmadja et al., 2020; de2), however these works focus primarily on small convolutional networks and struggle to scale to more typical deep networks. Other work has studied the limits of these convex relaxations on these small networks using vast amounts of CPU-compute (Salman et al., 2019). Recent SDP-based approaches (Dathathri et al., 2020) can produce much tighter bounds on these small networks.

**Lagrangian-based bounds** Related to our work is that which solves the Lagrangian of the LP relaxation (Dvijotham et al., 2018; Bunel et al., 2020a), which can tighten the bound but do not aim to solve the LP exactly due to relatively slow convergence. However, these works primarily study small networks whose LP relaxation can still be solved exactly with Gurobi. Although these works could in theory be used on larger networks, only the faster, linear-based bounds (Xu et al., 2020) have demonstrated applicability to standard deep learning architectures. In our work, we solve the LP relaxation exactly in *large* network settings that previously have only been studied with loose bounds of the LP relaxation such as LiRPA (Xu et al., 2020).

**Complete verification methods** These methods verify properties of deep networks exactly (without false positives or false negatives) using methods such as SMT solvers (Scheibler et al., 2015; Ehlers, 2017; Katz et al., 2017) and MILP solvers (Lomuscio & Maganti, 2017; Cheng et al., 2017; Dutta et al., 2018; Fischetti & Jo, 2018). Complete verification methods typically rely on BaB algorithms (Bunel et al., 2017), in which the verification problem is divided into subproblems (branching) that can be verified using incomplete verification methods (bounding) (Bunel et al., 2020b). However, these methods have a worst-case exponential runtime and have difficulty scaling beyond relatively small convolutional networks. Motivated by this, several recent works have improved the practical running time of BaB methods with custom solvers for medium-sized networks (Bunel et al., 2020a; De Palma et al., 2021; Xu et al., 2021; Wang et al., 2021), but have yet to scale to larger deep networks.

**Operator splitting methods** Operator splitting, and in particular the ADMM method, is a powerful technique in solving structured convex optimization problems and has applications in numerous settings ranging from optimal control (O'Donoghue et al., 2013) to training neural networks (Taylor et al., 2016). These methods scale well with the problem dimensions, can exploit sparsity in the problem data efficiently (Zheng et al., 2017), are amenable to parallelization on GPU (Schubiger et al., 2020), and have well-understood convergence properties under minimal regularity assumptions (Boyd et al., 2011). The benefit of ADMM as an alternative to interior-point solvers has been shown in various classes of optimization problems (O'donoghue et al., 2016). Our operator splitting method is specifically tailored for neural network verification in order to fully exploit the problem structure.

**Notation.** We denote the set of real numbers by $\mathbb{R}$, the set of nonnegative real numbers by $\mathbb{R}_+$, the set of real $n$-dimensional vectors by $\mathbb{R}^n$, the set of $m \times n$-dimensional matrices by $\mathbb{R}^{m \times n}$, and the $n$-dimensional identity matrix by $I_n$. The $p$-norm ($p \geq 1$) is denoted by $\| \cdot \|_p \colon \mathbb{R}^n \to \mathbb{R}_+$. For a set $\mathcal{S}$, we define the indicator function $\mathbb{I}_{\mathcal{S}}(x)$ of $\mathcal{S}$ as $\mathbb{I}_{\mathcal{S}}(x) = 0$ if $x \in \mathcal{S}$ and $\mathbb{I}_{\mathcal{S}}(x) = +\infty$ otherwise. Given a function $f \colon \mathcal{X} \to \mathcal{Y}$, the graph of $f$ is the set $\mathcal{G}_f = \{(x, f(x)) \mid x \in \mathcal{X}\}$.

## 2 Scalable Neural Network Verification via Operator Splitting

We consider an $\ell$-layer feed-forward neural network $f(x) \colon \mathbb{R}^{n_0} \to \mathbb{R}^{n_\ell}$ described by the following recursive equations,

$$x_0 = x, \quad x_{k+1} = \phi_k(x_k) \quad k = 0, \cdots, \ell - 1, \quad f(x) = x_\ell \tag{2}$$

where $x_0 \in \mathbb{R}^{n_0}$ is the input to the neural network, $x_k \in \mathbb{R}^{n_k}$ is the input to the $k$-th layer, and $\phi_k \colon \mathbb{R}^{n_k} \to \mathbb{R}^{n_{k+1}}$ is the operator of the $k$-th layer, which can represent any commonly-used operator in deep networks, such as linear (convolutional) layers, MaxPooling units, and activation functions.

Given the neural network $f$, a specification function $J \colon \mathbb{R}^{n_\ell} \mapsto \mathbb{R}$, and an input set $\mathcal{X} \subset \mathbb{R}^{n_0}$, we say that $f$ satisfies the specification $J$ if $J(f(x)) \geq 0$ for all $x \in \mathcal{X}$. This is equivalent to verifying that the optimal value of (1) is non-negative. We assume $\mathcal{X} \subset \mathbb{R}^{n_0}$ is a closed convex set and $J \colon \mathbb{R}^{n_\ell} \to \mathbb{R} \cup \{+\infty\}$ is a convex function, defining a performance measure on the output of the network. Note that our formulation generalizes to arbitrary computational graphs (e.g. residual blocks) and general objective functions (see Appendix B).

Using the sequential representation of the neural network in (2), we may rewrite the optimization problem in (1) as the following constrained optimization problem,

$$J^\star \leftarrow \text{minimize} \quad J(x_\ell) \quad \text{subject to} \quad x_{k+1} = \phi_k(x_k), \quad k = 0, \cdots, \ell - 1, \quad x_0 \in \mathcal{X}, \tag{3}$$

with $n := \sum_{k=0}^{\ell} n_k$ decision variables. We can rewrite (3) equivalently as

$$J^\star \leftarrow \text{minimize} \quad J(x_\ell) \quad \text{subject to} \quad (x_k, x_{k+1}) \in \mathcal{G}_{\phi_k}, \quad k = 0, \cdots, \ell - 1, \quad x_0 \in \mathcal{X}, \tag{4}$$

where $\mathcal{G}_{\phi_k} = \{(x_k, x_{k+1}) \mid x_{k+1} = \phi_k(x_k), \ \underline{x}_k \leq x_k \leq \bar{x}_k\}$ is the graph of $\phi_k$. Here $\underline{x}_k$ and $\bar{x}_k$ are *a priori* known bounds on $x_k$ when $x_0 \in \mathcal{X}$. The problem in (4) is non-convex due to presence of nonlinear operators in the network, such as activation layers. By over-approximating $\mathcal{G}_{\phi_k}$ by a convex set (or ideally by its convex hull), we arrive at a direct layer-wise convex relaxation of the problem. However, solving this relaxation directly cannot scale to even medium-sized neural networks Salman et al. (2019). In this section, by exploiting the sequential structure of the constraints, we propose a reformulation of (3) whose convex relaxation can be solved efficiently and in a scalable manner.

### 2.1 Variable Splitting

By introducing the intermediate variables $y_k$ and $z_k$, we can rewrite (3) as

$$
\begin{aligned}
J^\star \leftarrow \text{minimize} \quad & J(x_\ell) \\
\text{subject to} \quad & y_k = x_k, \ z_k = \phi_k(y_k), \ x_{k+1} = z_k, \quad k = 0, \cdots, \ell - 1, \\
& x_0 \in \mathcal{X}
\end{aligned} \tag{5}
$$

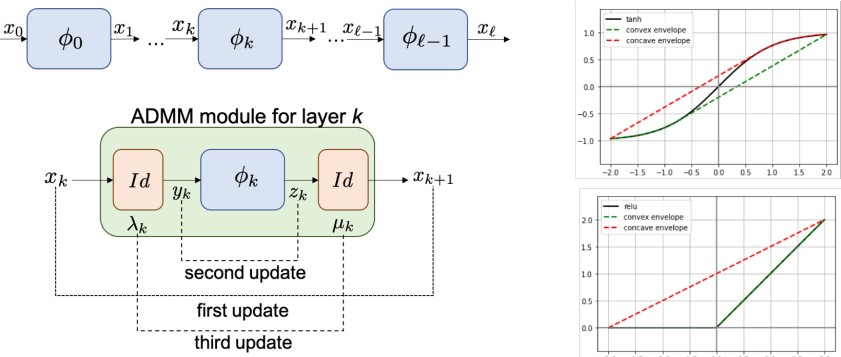

Figure 1: Left: Illustration of the network structure (top) and DeepSplit computation module for a generic layer (bottom). Adding identity layers in between the neural network layers decouples the variables $x_k$ and allows processing them independently. Right: Over-approximation of the graph of tanh (top) and ReLU (bottom) function by convex hull.

which has now $3n - n_0 - n_\ell$ decision variables. Intuitively, we have introduced additional "identity layers" between consecutive layers (see Figure 1). By overapproximating $\mathcal{G}_{\phi_k}$ by a convex set $\mathcal{S}_{\phi_k}$, we obtain the convex relaxation

$$
\begin{aligned}
J^\star_{\text{relaxed}} \leftarrow \text{minimize} \quad & J(x_\ell) \\
\text{subject to} \quad & y_k = x_k, \ (y_k, z_k) \in \mathcal{S}_{\phi_k}, \ x_{k+1} = z_k, \quad k = 0, \cdots, \ell - 1, \\
& x_0 \in \mathcal{X},
\end{aligned}
\tag{6}
$$

for which $J^\star_{\text{relaxed}} \leq J^\star$. This form is known as consensus as $y_k$ and $z_{k-1}$ are just copies of the variable $x_k$. As shown below, this "overparameterization" allows us to split the optimization problem into smaller sub-problems that can be solved in parallel and often in closed form.

## 2.2 Lagrangian Relaxation and Operator Splitting

We use $\mathbf{x} = (x_0, \cdots, x_\ell)$, $\mathbf{y} = (y_0, \cdots, y_{\ell-1})$ and $\mathbf{z} = (z_0, \cdots, z_{\ell-1})$ to denote the concatenated variables. By relaxing the equality constraints with Lagrangian multipliers, we define the augmented Lagrangian for (6) as follows,

$$
\mathcal{L}(\mathbf{x}, \mathbf{y}, \mathbf{z}, \boldsymbol{\lambda}, \boldsymbol{\mu}) = J(x_\ell) + \sum_{k=0}^{\ell-1} \mathbb{I}_{\mathcal{S}_{\phi_k}}(y_k, z_k) + \mathbb{I}_{\mathcal{X}}(x_0) + (\rho/2) \sum_{k=0}^{\ell-1} \left( \|x_k - y_k + \lambda_k\|_2^2 - \|\lambda_k\|_2^2 \right)
$$

$$
+ (\rho/2) \sum_{k=0}^{\ell-1} \left( \|x_{k+1} - z_k + \mu_k\|_2^2 - \|\mu_k\|_2^2 \right).
\tag{7}
$$

where $\boldsymbol{\lambda} = (\lambda_0, \cdots, \lambda_{\ell-1})$ and $\boldsymbol{\mu} = (\mu_0, \cdots, \mu_{\ell-1})$ are the scaled dual variables (by $1/\rho$) and $\rho > 0$ is the augmentation constant. Note that we have only relaxed the equality constraints in (6), and the constraints describing the sets $\mathcal{S}_{\phi_k}$ as well as $\mathcal{X}$ are kept intact. Furthermore, the inclusion of augmentation will render the dual function differentiable (see the Appendix for more details), and hence, easier to optimize. For the Lagrangian in (7), the dual function, which provides a lower bound to $J^\star_{\text{relaxed}}$, is given by $g(\boldsymbol{\lambda}, \boldsymbol{\mu}) = \inf_{(\mathbf{x}, \mathbf{y}, \mathbf{z})} \mathcal{L}(\mathbf{x}, \mathbf{y}, \mathbf{z}, \boldsymbol{\lambda}, \boldsymbol{\mu})$. The best lower bound can then be found by maximizing the dual function. [1]

Solving the inner problem jointly over $(\mathbf{x}, \mathbf{y}, \mathbf{z})$ to find the dual function is as difficult as solving a direct convex relaxation of (3). Instead, we split the primal variables $(\mathbf{x}, \mathbf{y}, \mathbf{z})$ into $\mathbf{x}$ and $(\mathbf{y}, \mathbf{z})$ and apply the classical ADMM algorithm to obtain the following iterations

---

[1]If strong duality holds, then this best lower bound would match $J^\star_{\text{relaxed}}$. As shown in Salman et al. (2019), strong duality holds under mild conditions.

---

**Algorithm 1:** DeepSplit Algorithm

---

**Data:** neural network $f$ (Eq. 2), bounded convex input set $\mathcal{X}$, convex function $J$.
**Result:** lower bound $J^{\star}_{\text{relaxed}}$ on Problem (1).
**Initialization:** $x_0 \in \mathcal{X}$, $x_{k+1} = \phi_k(x_k)$, $y_k = x_k$, $z_k = x_{k+1}$, $\lambda_k = 0, \mu_k = 0$,
$k = 0, \cdots, \ell - 1$, augmentation constant $\rho > 0$.
**repeat**
   |   **Step 1: x**-update (9)
   |   **Step 2: (y, z)**-update (10)
   |   **Step 3:** dual update (13)
**until** *stopping criterion is met*;
**Output:** $J(x_\ell)$

---

(shown in Figure 1) for updating the primal and dual variables,

$$\mathbf{x}^+ \in \text{argmin}_{\mathbf{x}} \ \mathcal{L}(\mathbf{x}, \mathbf{y}, \mathbf{z}, \boldsymbol{\lambda}, \boldsymbol{\mu}) \tag{8a}$$

$$(\mathbf{y}^+, \mathbf{z}^+) \in \text{argmin}_{(\mathbf{y}, \mathbf{z})} \ \mathcal{L}(\mathbf{x}^+, \mathbf{y}, \mathbf{z}, \boldsymbol{\lambda}, \boldsymbol{\mu}) \tag{8b}$$

$$(\boldsymbol{\lambda}^+, \boldsymbol{\mu}^+) = (\boldsymbol{\lambda}, \boldsymbol{\mu}) + \nabla_{(\boldsymbol{\lambda}, \boldsymbol{\mu})} \ \mathcal{L}(\mathbf{x}^+, \mathbf{y}^+, \mathbf{z}^+, \boldsymbol{\lambda}, \boldsymbol{\mu}). \tag{8c}$$

As we show below, the Lagrangian has a separable structure by construction that can be exploited in order to efficiently implement each step of (8).

### 2.3 THE x-UPDATE

The Lagrangian in (7) is separable across the $x_k$ variables; hence, the minimization in (8a) can be done independently for each $x_k$. Specifically, for $k = 0$, we obtain the following update rule for $x_0$,

$$x_0^+ = \text{Proj}_{\mathcal{X}}(y_0 - \lambda_0). \tag{9a}$$

Projections onto the $\ell_\infty$ and $\ell_2$ balls can be done in closed-form. For the $\ell_1$ ball, we can use the efficient projection scheme proposed in (Duchi et al., 2008), which has $\mathcal{O}(n_0)$ complexity in expectation. For subsequent layers $k = 1, \cdots, \ell$, we obtain the updates

$$x_k^+ = \frac{1}{2}(y_k - \lambda_k + z_{k-1} - \mu_{k-1}) \tag{9b}$$

$$x_\ell^+ = \arg\min_{x_\ell} \ J(x_\ell) + \frac{\rho}{2}\|x_\ell - z_{\ell-1} + \mu_{\ell-1}\|_2^2. \tag{9c}$$

For convex $J$ and $\rho > 0$, the optimization problem for updating $x_\ell$ is strongly convex with a unique optimal solution. Indeed, its solution is the proximal operator of $J/\rho$ evaluated at $z_{\ell-1} - \mu_{\ell-1}$. For the special case of linear objectives, $J(x_\ell) = c^\top x_\ell$, we obtain the closed-form solution $x_\ell^+ = -c/\rho + (z_{\ell-1} - \mu_{\ell-1})$.

### 2.4 THE (y, z)-UPDATE

Similarly, the Lagrangian is also separable across the $(y_k, z_k)$ variables. Updating these variables in (8b) corresponds to the following projection operations per layer,

$$(y_k^+, z_k^+) = \text{Proj}_{\mathcal{S}_{\phi_k}}(x_k^+ + \lambda_k, x_{k+1}^+ + \mu_k), \tag{10}$$

for $k = 0, \cdots, \ell - 1$. Depending on the type of the layer (linear, activation, convolution, etc.), we obtain different projections, which we describe below.

**Affine Layers.** Suppose $\phi_k(y_k) = W_k y_k + b_k$ is an affine layer representing a fully-connected, convolutional, or an average pooling layer. Then the graph of $\phi_k$ is already a convex set given by $\mathcal{G}_{\phi_k} = \{(y_k, z_k) \mid z_k = W_k y_k + b_k\}$. Choosing $\mathcal{S}_{\phi_k} = \mathcal{G}_{\phi_k}$, the projection in (10) takes the form

$$\begin{aligned} y_k^+ &= (I_{n_k} + W_k^\top W_k)^{-1}(x_k^+ + \lambda_k + W_k^\top(x_{k+1}^+ + \mu_k - b_k)) \\ z_k^+ &= W_k y_k^+ + b_k. \end{aligned} \tag{11}$$

The matrix $(I_{n_k} + W_k^\top W_k)^{-1}$ can be pre-computed and cached for subsequent iterations. We can do this efficiently for convolutional layers using the fast Fourier transform, which we discuss later in Section D.3.

**Activation Layers.** For an activation layer of the form $\phi(x) := [\varphi_1(x_1) \; \cdots \; \varphi_n(x_n)]^\top$, the convex relaxation of $\mathcal{G}_\phi$ is given by the Cartesian product of individual convex relaxations i.e., $\mathcal{S}_\phi = \mathcal{S}_{\varphi_1} \times \cdots \times \mathcal{S}_{\varphi_n}$. For a generic activation function $\varphi \colon \mathbb{R} \to \mathbb{R}$, suppose we have a concave upper bound $\bar{\varphi}$ and a convex lower bound $\underline{\varphi}$ on $\varphi$ over an interval $I = [\underline{x}, \bar{x}]$, i.e., $\underline{\varphi}(x) \leq \varphi(x) \leq \bar{\varphi}(x) \; \forall \; x \in [\underline{x}, \bar{x}]$. A convex overapproximation of $\mathcal{G}_\varphi$ is

$$\mathcal{S}_\varphi = \{(x, y) \mid \underline{\varphi}(x) \leq y \leq \bar{\varphi}(x), \; \underline{x} \leq x \leq \bar{x}\}, \tag{12}$$

which turns out to be the convex hull of $\mathcal{G}(\varphi)$ when $\bar{\varphi}$ and $\underline{\varphi}$ are concave and convex envelopes of $\varphi$, respectively. As an example, when $\underline{x} < 0 < \bar{x}$, the ReLU activation function $y = \max(0, x)$ admits the envelopes $\underline{\varphi}(x) = \max(0, x)$, $\bar{\varphi}(x) = \underline{y} + \frac{\bar{y} - \underline{y}}{\bar{x} - \underline{x}}(x - \underline{x})$ on $[\underline{x}, \bar{x}]$, where $\underline{y} = \max(0, \underline{x})$ and $\bar{y} = \max(0, \bar{x})$ (Ehlers, 2017; Wong & Kolter, 2017) (see Figure 1). The assumed pre-activation bounds $\underline{x}$ and $\bar{x}$ used to relax the activation functions can be obtained a priori via a variety of existing techniques (Wong & Kolter, 2017; Zhang et al., 2018; 2019). We present projection operators of additional layers in Appendix C.

## 2.5 The $(\boldsymbol{\lambda}, \boldsymbol{\mu})$-update

Finally, we update the scaled dual variables as follows,

$$\begin{aligned}
\lambda_k^+ &= \lambda_k + (x_k^+ - y_k^+), & k &= 0, \cdots, \ell - 1, \\
\mu_k^+ &= \mu_k + (x_{k+1}^+ - z_k^+), & k &= 0, \cdots, \ell - 1.
\end{aligned} \tag{13}$$

The DeepSplit Algorithm is summarized in Algorithm 1. We provide further details such as parameter selection and efficient implementations for convolutional layers in Appendix D.

## 3 Connection to Lagrangian-based methods

In this section, we draw connections to two related methods relying on Lagrangian relaxation. Specifically, we relate our approach to an earlier dual method (Dvijotham et al., 2018) (Section 3.1), as well as a recent Lagrangian decomposition method (Bunel et al., 2020a) (Section 3.2). Overall, these approaches use a similar Lagrangian formulation, but the specific choices in splitting and augmentation of the Lagrangian result in poorer theoretical convergence guarantees when solving the convex relaxation to optimality.

### 3.1 Dual method via Lagrangian Relaxation of the nonconvex problem

Instead of splitting the neural network equations (3) with auxiliary variables, an alternative strategy is to directly relax (3) with Lagrangian multipliers (Dvijotham et al., 2018):

$$\mathcal{L}(\mathbf{x}, \boldsymbol{\lambda}) = J(x_\ell) + \sum_{k=0}^{\ell-1} \lambda_k^\top (x_{k+1} - \phi_k(x_k)) + \mathbb{I}_{\mathcal{X}}(x_0). \tag{14}$$

where $\mathbf{x} = (x_0, \cdots, x_\ell)$ and $\boldsymbol{\lambda} = (\lambda_0, \cdots, \lambda_{\ell-1})$. This results in the dual problem $g^\star \leftarrow$ maximize $g(\boldsymbol{\lambda})$, where the dual function is

$$g(\boldsymbol{\lambda}) = \inf_{\underline{x}_\ell \leq x_\ell \leq \bar{x}_\ell} \{J(x_\ell) + \lambda_{\ell-1}^\top x_\ell\} + \sum_{k=1}^{\ell-1} \inf_{\underline{x}_k \leq x_k \leq \bar{x}_k} \{\lambda_{k-1}^\top x_k - \lambda_k^\top \phi_k(x_k)\} + \inf_{x_0 \in \mathcal{X}_0} \{-\lambda_0^\top \phi_0(x_0)\}.$$

By weak duality, $g^\star \leq J^\star$. The inner minimization problems in $g(\boldsymbol{\lambda})$ to compute the dual function for a given $\boldsymbol{\lambda}$ can often be solved efficiently or even in closed-form (Dvijotham et al., 2018). The resulting dual problem is unconstrained but non-differentiable; hence it is solved using dual subgradient method (Dvijotham et al., 2018). However, subgradient methods are known to be very slow with convergence rate $O(1/\sqrt{N})$ where $N$ is the number of updates (Nesterov, 2003), making it inefficient to find exact solutions to the convex relaxation. On the other hand, this method can be stopped at any time to obtain a valid lower bound.

### 3.2 Lagrangian method via a non-separable splitting

Another related approach is the Lagrangian decomposition method from Bunel et al. (2020a). To decouple the constraints for the convex relaxation of (3), this approach can be viewed as introducing *one* set of intermediate variables $y_k$ as copies of $x_k$ to obtain

$$
\begin{aligned}
J^{\star}_{\text{relaxed}} \leftarrow \text{minimize} \quad & J(y_\ell) \\
\text{subject to} \quad & (y_k, x_{k+1}) \in \mathcal{S}_{\phi_k} \quad k = 0, \cdots, \ell - 1 \\
& y_k = x_k \qquad\qquad k = 0, \cdots, \ell \\
& x_0 \in \mathcal{X}
\end{aligned}
\tag{15}
$$

This splitting is in the spirit of the splitting introduced in Bunel et al. (2020a;b),[2] and differs from our splitting which uses *two* sets of variables in (5). By relaxing the consensus constraints $y_k = x_k$, the Lagrangian is

$$
\mathcal{L}(\mathbf{x}, \mathbf{y}, \boldsymbol{\mu}) = J(y_\ell) + \sum_{k=0}^{\ell} \mu_k^\top (y_k - x_k) + \sum_{k=0}^{\ell-1} \mathbb{I}_{\mathcal{S}_{\phi_k}}(y_k, x_{k+1}) + \mathbb{I}_{\mathcal{X}}(x_0).
\tag{16}
$$

Again the Lagrangian is separable and its minimization results in the following dual function

$$
g(\boldsymbol{\mu}) = \inf_{\underline{x}_\ell \leq y_\ell \leq \bar{x}_\ell} \left\{ J(y_\ell) + \mu_\ell^\top y_\ell \right\} + \sum_{k=0}^{\ell-1} \inf_{(y_k, x_{k+1}) \in \mathcal{S}_{\phi_k}} \left\{ \mu_k^\top y_k - \mu_{k+1}^\top x_{k+1} \right\} + \inf_{x_0 \in \mathcal{X}} \left\{ -\mu_0^\top x_0 \right\}.
$$

Since the dual function is not differentiable, it must be maximized by a subgradient method, which again has an $O(1/\sqrt{N})$ rate. To induce differentiability in the dual function and improve speed, Bunel et al. (2020a) uses Augmented Lagrangian (AL). Since only one set of variables was introduced in (15), the AL is no longer separable across the primal variables. Therefore, for each update of the dual variable, the AL must be minimized iteratively. To this end, Bunel et al. (2020a) use the Frank-Wolfe Algorithm in a block-coordinate as an iterative subroutine. However, this slows down overall convergence and suffers from compounding errors when the sub-problems are not fully solved. When stopping early, the primal minimization must be solved to convergence in order to compute the dual function and produce a valid bound.

In contrast to the approach described above, in this paper we used a different variable splitting scheme in (5) that allows us to *fully* separate layers in a neural network. This subtle difference has a significant impact: we can efficiently minimize the corresponding AL in *closed form*, without resorting to any iterative subroutine. Specifically, we use the ADMM algorithm, which is known to converge at an $O(1/N)$ rate (He & Yuan, 2012). In summary, our method enjoys an order of magnitude faster theoretical convergence, is more robust to numerical errors, and has minimal requirements for parameter tuning. To stop early, we can use a similar strategy as the Frank-Wolfe approach from Bunel et al. (2020a) and run the primal iteration to convergence with fixed dual variables to get a valid bound.

## 4 Experiments

The strengths of our method are (a) its ability to exactly solve LP relaxations and (b) do so at scales that were previously impossible. To evaluate this, we first demonstrate how solving the LP to optimality leads to tighter certified robustness guarantees in image classification and reinforcement learning tasks (Section 4.1). We then stress test our method in both speed and scalability against a commercial LP solver and in the large network setting (Section 4.2).

**Setup** In all the experiments, we focus on the setting of *verification-agnostic* networks, similar to Dathathri et al. (2020). However, we focus on significantly *larger* networks that

---

[2]If we define $\phi_k(x_k) = W_{k+1}\sigma(x_k) + b_{k+1}$, where $W_{k+1}, b_{k+1}$ are the parameters of the affine layer and $\sigma$ is a layer of activation functions, this splitting coincides with the one proposed in Bunel et al. (2020a;b)

Table 1: Certified test accuracy (%) of PGD-trained models on CIFAR10 through ADMM, the Lagrangian decomposition methods (Dvijotham et al., 2018; Bunel et al., 2020b), and fast dual/linear (Wong et al., 2018; Xu et al., 2020) or interval bounds (Gowal et al., 2018). All the methods are given the same time budget.

| | Exact | Lagrangian methods | | | | Fast bounds | |
|---|---|---|---|---|---|---|---|
| $\epsilon$ | ADMM | Adam | Prox | Dual Adam | Dual Decomp Adam | Linear | IBP |
| 1/255 | **64.0** | 47.4 | 61.1 | 47.6 | 47.8 | 59.8 | 42.8 |
| 1.5/255 | **45.7** | 28.4 | 41.8 | 28.5 | 28.8 | 36.8 | 16.8 |
| 2/255 | **19.5** | 10.9 | 17.4 | 11.0 | 11.1 | 13.2 | 3.6 |
| 2.5/255 | **5.5** | 2.6 | 4.5 | 2.7 | 2.7 | 3.3 | 0.7 |

have previously only been feasibly bounded with fast linear-based bounds of the LP relaxation (Wong et al., 2018; Xu et al., 2020). All the networks have been trained adversarially with the PGD attack (Madry et al., 2017). Full details about the network architecture, network training, and the parameters for our splitting method can be found in Appendix F.

### 4.1 Improved Bounds From Exact LP Solutions with ADMM

We first demonstrate how solving the LP exactly with our method results in tighter bounds than prior work. We consider two main settings: certifying the robustness of classifiers for CIFAR10 and deep Q-networks (DQNs) in Atari games. We defer additional analogous results in the smaller MNIST setting to Appendix F.3.

**CIFAR10** Much work studying verification in CIFAR10 has focused primarily on a small CNN with only 6k hidden units—smaller than the classic LeNet from MNIST (LeCun et al., 2015). In this regime, tighter approaches such as SDP and branch-and-bound are applicable.

Instead, we consider an order of magnitude larger model, which cannot be feasibly solved by these alternatives. Up until this point, the only solutions for large networks were linear-based bounds to the LP relaxation (Wong et al., 2018; Xu et al., 2020). However, these bounds are known to be quite loose. How much better can we do if we solve the LP exactly?

In Table 1, we find that solving the LP exactly leads to significant gains in certified robustness for large networks, with up to 4% additional certified robustness over the best-performing alternative. All the methods in Table 1 are given the same time budget. Indeed, the better theoretical convergence guarantees of ADMM translate to better results in practice: when given a similar budget, the Lagrangian baselines have worse convergence and cannot verify as many examples. A complete description of these baselines is in Appendix F.3, with implementation details for the Lagrangian methods in Appendix G.

**State-robust RL** We demonstrate our approach on a non-classification benchmark from reinforcement learning: verifying the robustness of deep Q-networks (DQNs) to adversarial state perturbations (Zhang et al., 2020). Specifically, we verify whether a learned DQN policy outputs stable actions in the discrete space when given perturbed states. Similar to the large network considered in the CIFAR10 setting, this benchmark has only been demonstrably verified with fast but loose linear-based bounds (Xu et al., 2020).

Similar to the CIFAR10 setting, we observe consistent improvement in certified robustness of the DQN when solving the LP exactly with ADMM across multiple RL settings. We summarize the results using our method and the linear-based bounds on LP relaxations (Wong et al., 2018; Xu et al., 2020) in Table 2. Further details regarding the dataset, the DQN architectures, and this task can be found in Appendix F.4.

### 4.2 Speed and Scalability

In this section, we stress test several aspects of DeepSplit for solving LP relaxations. We first compare solving speeds of our ADMM-based LP solver to a commercial-grade LP solver. We then push the limits of architecture size and solve the LP relaxation for a standard ResNet18.

Table 2: The percentage of actions from a deep Q-network that are certifiably robust to changes in the state space for three RL tasks: Bankheist, Roadrunner, and Pong. We compare fast linear bounds (Linear) (Wong et al., 2018; Xu et al., 2020) and ADMM.

| | Bankheist | | | Roadrunner | | | Pong | |
|---|---|---|---|---|---|---|---|---|
| $\epsilon$ | Linear | ADMM | $\epsilon$ | Linear | ADMM | $\epsilon$ | Linear | ADMM |
| 0.0016 | 67.0 | **71.4** | 0.0012 | 32.6 | **36.6** | 0.0004 | 96.1 | **97.4** |
| 0.0020 | 39.7 | **49.5** | 0.0016 | 26.3 | **27.5** | 0.0008 | 93.4 | **95.6** |
| 0.0024 | 12.7 | **25.9** | 0.0020 | 19.6 | **22.8** | 0.0012 | 92.1 | **94.3** |
| 0.0027 | 1.4 | **7.3** | 0.0024 | 1.1 | **3.7** | 0.0016 | 82.1 | **84.0** |

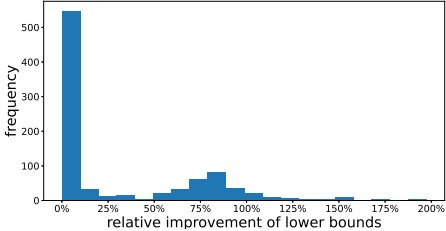 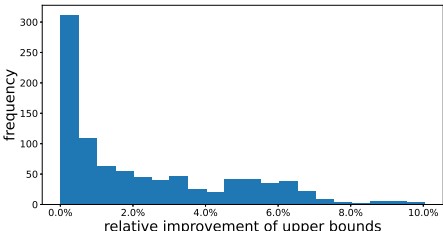

Figure 2: A total of 1000 ResNet18 output lower and upper bounds are computed from ADMM and LiRPA for comparison in CIFAR10. Histograms of the relative improvement percentage of ADMM over LiRPA are shown for the lower (left) and upper (right) bounds, which have an average relative improvement of 31.61% and 2.32%, respectively.

**Speed comparison**    We first compare the solving speeds of our method with state-of-the-art solvers for convex relaxations: a commercial-grade LP solver, Gurobi. Since Gurobi cannot handle large networks, we benchmark the approaches on a fully connected network that Gurobi can handle (see Appendix H for further details on experimental setup, as well as additional speed comparisons to alternative approaches).

We find that our method provides a nearly 7x speedup over Gurobi. On average, it takes ADMM 38 seconds per example to calculate these intermediate bounds using a single GeForce GTX 1080, in comparison to 258 seconds per example for Gurobi on an Intel Core i7-6700K.

**Scalability**    Finally, to test the scalability and generality of our approach, we consider solving the LP relaxation for a ResNet18, which up to this point has simply not been possible. The only applicable method here is LiRPA (Xu et al., 2020)—a highly scalable implementation of the linear-based bounds that works for arbitrary networks but can be quite loose in practice. We defer specific experimental details to Appendix I.

We find that exact LP solutions with our ADMM solver can also produce substantial improvements in the bound at ResNet18 scales, as shown in Figure 2. For a substantial number of examples, we find that ADMM can find significantly tighter bounds (especially for lower bounds). A tabular presentation of the results is in Table 5 of Appendix I.

## 5    CONCLUSION

In this paper, we proposed DeepSplit, a scalable and modular operator splitting technique for solving convex relaxation-based verification problems for neural networks. The method can exactly solve large-scale LP relaxations with GPU acceleration with favorable convergence rates. Our approach leads to significantly tighter bounds across a range of classification and reinforcement learning benchmarks, and can scale to a standard ResNet18. We leave as future work a further investigation of variations of ADMM that can improve convergence rates in deep learning-sized problem instances, as well as extensions beyond the LP setting.

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

Demonstration of our codes are anonymously available in the attachment.

## A    CONVERGENCE ANALYSIS OF DEEPSPLIT

By defining $\mathbf{x}_1 = \mathbf{x}$, $\mathbf{x}_2 = (\mathbf{y}, \mathbf{z})$ (the primal variables) and $\boldsymbol{\nu} = (\boldsymbol{\lambda}, \boldsymbol{\mu})$ (the scaled dual variables), we can write the split convex relaxation in (6) as

$$\text{minimize} \quad f_1(\mathbf{x}_1) + f_2(\mathbf{x}_2) \tag{17}$$
$$\text{subject to } A_1\mathbf{x}_1 + A_2\mathbf{x}_2 = 0$$

with the corresponding Augmented Lagrangian

$$\mathcal{L}(\mathbf{x}_1, \mathbf{x}_2, \boldsymbol{\nu}) = f_1(\mathbf{x}) + f_2(\mathbf{x}_2) + \frac{\rho}{2}(\|A_1\mathbf{x}_1 + A_2\mathbf{x}_2 + \boldsymbol{\nu}\|_2^2 - \|\boldsymbol{\nu}\|_2^2) \tag{18}$$

where $f_1 \colon \mathbb{R}^n \to \mathbb{R} \cup \{+\infty\}$ and $f_2 \colon \mathbb{R}^{2n-2n_\ell} \to \mathbb{R} \cup \{+\infty\}$ are extended real-valued functions defined as $f_1(\mathbf{x}_1) := J(x_\ell) + \mathbb{I}_{\mathcal{X}}(x_0)$, $f_2(\mathbf{x}_2) := \sum_{k=0}^{\ell-1} \mathbb{I}_{\mathcal{S}_{\phi_k}}(y_k, z_k)$. Moreover, $A_1\mathbf{x}_1 + A_2\mathbf{x}_2 = 0$ represents the set of equality constraints $y_k = x_k$ and $x_{k+1} = z_k$ for $k = 0, \cdots, \ell$. The dual function is given by

$$g(\boldsymbol{\nu}) = \inf_{\mathbf{x}_1, \mathbf{x}_2} \mathcal{L}(\mathbf{x}_1, \mathbf{x}_2, \boldsymbol{\nu}). \tag{19}$$

By Danskin's theorem (Bertsekas, 1997), the sub-differential of the dual function is given by

$$\partial g(\boldsymbol{\nu}) = \{A_1\bar{\mathbf{x}}_1 + A_2\bar{\mathbf{x}}_2 \colon (\bar{\mathbf{x}}_1, \bar{\mathbf{x}}_2) \in \arg\min \mathcal{L}(\mathbf{x}_1, \mathbf{x}_2, \boldsymbol{\nu})\} \tag{20}$$

Here $\bar{\mathbf{x}}_1, \bar{\mathbf{x}}_2$ are minimizers of the Lagrangian (not necessarily unique), which satisfy the optimality conditions

$$0 \in \partial f_1(\bar{\mathbf{x}}_1) + \rho A_1^\top(A_1\bar{\mathbf{x}}_1 + A_2\bar{\mathbf{x}}_2 + \boldsymbol{\nu}) \tag{21}$$
$$0 \in \partial f_2(\bar{\mathbf{x}}_2) + \rho A_2^\top(A_1\bar{\mathbf{x}}_1 + A_2\bar{\mathbf{x}}_2 + \boldsymbol{\nu})$$

First, we show that the sub-differential is a singleton, i.e., $g$ is continuously differentiable. Suppose $(\bar{\mathbf{x}}_1, \bar{\mathbf{x}}_2) \in \arg\min \mathcal{L}(\boldsymbol{\xi}, \boldsymbol{\nu})$ and $(\bar{\mathbf{w}}_1, \bar{\mathbf{w}}_2) \in \arg\min \mathcal{L}(\boldsymbol{\xi}, \boldsymbol{\nu})$ are two distinct minimizers of the Lagrangian, hence satisfying

$$0 \in \partial f_1(\bar{\mathbf{w}}_1) + \rho A_1^\top(A_1\bar{\mathbf{w}}_1 + A_2\bar{\mathbf{w}}_2 + \boldsymbol{\nu}) \tag{22}$$
$$0 \in \partial f_2(\bar{\mathbf{w}}_2) + \rho A_2^\top(A_1\bar{\mathbf{w}}_1 + A_2\bar{\mathbf{w}}_2 + \boldsymbol{\nu})$$

By monotonicity of the sub-differentials, we can write

$$(\partial f_1(\bar{\mathbf{x}}_1) - \partial f_1(\bar{\mathbf{w}}_1))^\top(\bar{\mathbf{x}}_1 - \bar{\mathbf{w}}_1) \geq 0 \tag{23}$$
$$(\partial f_2(\bar{\mathbf{x}}_2) - \partial f_2(\bar{\mathbf{w}}_2))^\top(\bar{\mathbf{x}}_2 - \bar{\mathbf{w}}_2) \geq 0$$

By substituting (21) and (22) in (23), we obtain

$$-(\rho A_1^\top(A_1\bar{\mathbf{x}}_1 + A_2\bar{\mathbf{x}}_2 + \boldsymbol{\nu}) - \rho A_1^\top(A_1\bar{\mathbf{w}}_1 + A_2\bar{\mathbf{w}}_2 + \boldsymbol{\nu}))^\top(\bar{\mathbf{x}}_1 - \bar{\mathbf{w}}_1) \geq 0 \tag{24}$$
$$-(\rho A_2^\top(A_1\bar{\mathbf{x}}_1 + A_2\bar{\mathbf{x}}_2 + \boldsymbol{\nu}) - \rho A_2^\top(A_1\bar{\mathbf{w}}_1 + A_2\bar{\mathbf{w}}_2 + \boldsymbol{\nu}))^\top(\bar{\mathbf{x}}_2 - \bar{\mathbf{w}}_2) \geq 0$$

By adding the preceding inequalities, we obtain

$$-\rho\|A_1\bar{\mathbf{x}}_1 + A_2\bar{\mathbf{x}}_2 - (A_1\bar{\mathbf{w}}_1 + A_2\bar{\mathbf{w}}_2)\|_2^2 \geq 0 \tag{25}$$

This implies that $A_1\bar{\mathbf{x}}_1 + A_2\bar{\mathbf{x}}_2 = A_1\bar{\mathbf{w}}_1 + A_2\bar{\mathbf{w}}_2$ and hence, the sub-differential $\partial g$ is a singleton.

*Convergence.* When $\mathcal{X}$ is a closed nonempty convex set, $\mathbb{I}_{\mathcal{X}}(x_0)$ is a convex closed proper (CCP) function. Assuming that $J$ is also CCP, then we can conclude that $f_1$ is CCP. Furthermore, since the sets $\mathcal{S}_{\phi_k}$ are nonempty convex sets, we can conclude that $f_2$ is CCP. Under these assumptions, the augmented Lagrangian has a minimizer (not necessarily unique) for each value of the dual variables. Finally, under the assumption that the Augmented Lagrangian has a saddle point (which produces a solution to (17)), the ADMM algorithm we have primal convergence $\|r_p\| \to 0$ (see (31)), dual residual convergence $\|r_d\| \to 0$, as well as objective convergence $J(x_\ell) \to J^\star$ (Boyd et al., 2011).

We remark that the convergence guarantees of ADMM holds even if $f_1$ and $f_2$ assume the value $+\infty$. This is the case for indicator functions resulting in projections in the first two updates of ADMM.

# B  ADMM FOR GENERAL COMPUTATIONAL GRAPHS

To handle general computational graphs, we follow the approach used in (Wong et al., 2018). Specifically, consider a generalized $\ell$-"layer" neural network given by the equations

$$x_{k+1} = \sum_{i=1}^{k} \phi_{ik}(x_i) \equiv \Phi_k(x_{1:k}) \quad k = 0, \cdots, \ell - 1 \tag{26}$$

where $x_{1:k}$ denotes the concatenated vector of variables $(x_1, \cdots, x_k)$. Note that this view of a network reduces to the feedforward equations in (2) when $\phi_{ik}(x_i) = 0$ for all $i < k$. Furthermore, this view subsumes arbitrary skip connections, such as residual connections. Then, the graph form of the verification problem in (4) can be rewritten for the generalized neural network as

$$\begin{aligned} J^\star \leftarrow \text{minimize} \quad & J(x_\ell) \\ \text{subject to} \quad & (x_{1:k}, x_{k+1}) \in \mathcal{G}_{\phi_k(x_{1:k})} \quad k = 0, \cdots, \ell - 1 \\ & x_0 \in \mathcal{X}. \end{aligned} \tag{27}$$

Similar to before, let $S_{\phi_k}$ be a sound convex approximation of $\mathcal{G}_{\phi_k}$. Then, our convex relaxation introduces additional intermediate variables $y_{1:k}$ and $z_k$ to get the following convex relaxation (analogous to (6)):

$$\begin{aligned} p^\star_{\text{relaxed}} \leftarrow \text{minimize} \quad & J(x_\ell) \\ \text{subject to} \quad & y_{1:k} = x_{1:k} & k = 0, \cdots, \ell - 1 \\ & (y_{1:k}, z_k) \in \mathcal{S}_{\phi_k} & k = 0, \cdots, \ell - 1 \\ & x_{k+1} = z_k & k = 0, \cdots, \ell - 1 \\ & x_0 \in \mathcal{X}. \end{aligned} \tag{28}$$

Thus, producing an ADMM layer for a network with arbitrary skip connections reduces to deriving the corresponding projection operator onto $\mathcal{S}_{\phi_k}$. Since most residual connections simply add an affine transformation of a previous layer, these projections are straightforward and have a closed form (an example of a typical residual ADMM block is in Appendix C).

# C  ADMM LAYERS

## C.1  ReLU PROJECTION

Consider the ReLU function $y = \varphi(x) = \max(0, x)$ on the interval $x \in [\underline{x}, \bar{x}]$. The projection of a point $(x^{(0)}, y^{(0)})$ onto the convex hull of $G(\varphi)$ has a closed-form solution. We consider three cases.

1. Active neuron ($\underline{x} \geq 0$). In this case, the convex hull is given by
$$\mathcal{S}_\varphi = \{(x, y) \mid y = x, \ \underline{x} \leq x \leq \bar{x}\}.$$
and the projection of $(x^{(0)}, y^{(0)})$ onto $\mathcal{S}_\varphi$ is given by $x' = y' = \min(\max(\frac{x^{(0)} + y^{(0)}}{2}, \underline{x}), \bar{x})$.

2. Inactive neuron ($\bar{x} \leq 0$). In this case, the convex hull is
$$\mathcal{S}_\varphi = \{(x, y) \mid y = 0, \ \underline{x} \leq x \leq \bar{x}\}.$$
and the projection of $(x^{(0)}, y^{(0)})$ onto $\mathcal{S}_\varphi$ is given by $x' = \min(\max(x^{(0)}, \underline{x}), \bar{x}), \ y' = 0$.

3. Unknown neuron ($\underline{x} \leq 0 \leq \bar{x}$). The convex hull is given by
$$\mathcal{S}_\varphi = \{(x, y) \mid y \geq 0, \ y \geq x, \ y \leq \underline{y} + s(x - \underline{x})\}$$
where $s = \frac{\bar{y} - \underline{y}}{\bar{x} - \underline{x}}$, $\underline{y} = \max(0, \underline{x})$, $\bar{y} = \max(0, \bar{x})$. We first project $(x^{(0)}, y^{(0)})$ onto each facet of the triangle and then select the point with the minimal distance.

$$x^{(1)} = \min(\max(\frac{x^{(0)} + y^{(0)}}{2}, 0), \bar{x}) \quad y^{(1)} = x^{(1)}$$

$$x^{(2)} = \min(\max(\frac{x^{(0)} + sy^{(0)} + s(s\underline{x} - \underline{y})}{s^2 + 1}, \underline{x}), \bar{x}) \quad y^{(2)} = \frac{s(x^{(0)} - \underline{x}) + s^2 y^{(0)} + \underline{y}}{s^2 + 1}$$

$$x^{(3)} = \min(\max(0, x^{(0)}), \underline{x}) \quad y^{(3)} = 0$$

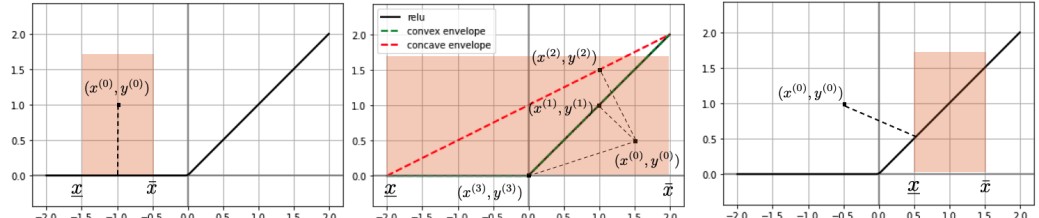

Figure 3: Projection of a point $(x_0, y_0)$ onto the convex hull of the ReLU function $y = \max(0, x)$ over the interval $[\underline{x}, \bar{x}]$. We first project $(x_0, y_0)$ onto all facets of the convex hull and then select the point with minimal distance to $(x_0, y_0)$.

The projected point is $(x', y') = (x^{(i^\star)}, y^{(i^\star)})$, where $i^\star = \arg\min_{1 \leq i \leq 3} \sqrt{(x^{(0)} - x^{(i)})^2 + (y^{(0)} - y^{(i)})^2}$.

## C.2 Convolutions

Although a convolution is technically a linear operator, it is impractical to directly form the inverse matrix from the projection step of the DeepSplit algorithm. Instead, we represent a typical convolutional layer $f_{\mathrm{conv}}$ with stride, padding and bias as

$$f_{\mathrm{conv}} = f_{\mathrm{bias}} \circ f_{\mathrm{ds}} \circ f_{\mathrm{crop}} \circ f_{\mathrm{circ}} \circ f_{pad} = f_{\mathrm{post}} \circ f_{\mathrm{circ}} \circ f_{\mathrm{pad}}, \tag{29}$$

where $f_{\mathrm{pad}}$ is a padding step, $f_{\mathrm{circ}}$ is a circular convolution step, $f_{\mathrm{crop}}$ is a cropping step, $f_{\mathrm{ds}}$ is a downsampling step to handle stride greater than one, and $f_{\mathrm{bias}}$ is a step that adds the bias. Then, the last three steps are combined into one post-processing step $f_{\mathrm{post}}$ to reduce the number of concensus constraints in the DeepSplit algorithm. The projection steps for all of these operators are presented next in Sections C.3 through C.6, and an efficient implementation of the projection step for the affine layer $f_{\mathrm{circ}}$ is in Appendix E.

## C.3 Padding

The padding layer $f_{\mathrm{pad}}$ takes an image as input and adds padding to it. Denote the input image by $y_k$ and the padded image by $z_k$. We can decompose the output $z_k$ into two vectors, $z_k^0$ which is a copy of the input $y_k$, and $z_k^1$ which represents the padded zeros on the edges of image. Equivalently, the padding layer $z_k = \phi_k(y_k)$ can be written in an affine form

$$z_k = \begin{bmatrix} z_k^0 \\ z_k^1 \end{bmatrix} = \begin{bmatrix} I \\ 0 \end{bmatrix} y_k = W_k y_k,$$

for which the projection operator reduces to the affine case.

## C.4 Cropping

The cropping layer $f_{\mathrm{crop}}$ crops the output of the circular convolution $f_{\mathrm{circ}}$ to the original size of the input image before padding. Denote $y_k$ the input image and $z_k$ the output image of the cropping layer. By decomposing the input image $y_k$ into the uncropped pixels $y_k^0$ and the cropped pixels $y_k^1$, the cropping layer $z_k = \phi_k(y_k)$ has an affine formulation

$$z_k = y_k^0 = \begin{bmatrix} I & 0 \end{bmatrix} \begin{bmatrix} y_k^0 \\ y_k^1 \end{bmatrix} = W_k y_k$$

whose projection operator is given in Section 2.4.

## C.5 Down-sampling and bias

If the typical convolutional layer $f_{\mathrm{conv}}$ has stride greater than one, a down-sampling layer is added in the DeepSplit algorithm, which essentially has the same affine form as the

cropping layer with different values of $y_k^0$ and $y_k^1$. Therefore, the projection operator for the down-sampling layer reduces to the affine case as well.

The bias layer in the DeepSplit algorithm handles the case when the convolutional layer $f_{\text{conv}}$ has a bias $b_k$ and is implemented by $z_k = \phi_k(y_k) = y_k + b_k$. This is an affine expression and its projection operator is given in Section 2.4.

### C.6 Convolutional post-processing layer

We combine the cropping, down-sampling and bias layers into one post-processing layer, i.e., $f_{\text{post}} = f_{\text{bias}} \circ f_{\text{ds}} \circ f_{\text{crop}}$, as shown in (29). This reduces the total number of concensus constraints in the DeepSplit algorithm. Since all the three layers are in fact affine, the post-processing layer is also affine and its projection operator can be obtained correspondingly.

### C.7 Maxpooling

Consider a Maxpooling layer $\phi(x) = \max_k x_k$, $k = 1, \cdots, n$ over $\underline{x} \leq x \leq \bar{x}$. (Bunel et al., 2020b, Appendix B.2) shows that the Maxpooling layer can be equivalently decomposed as a combination of linear and ReLU layers. This is motivated by the following fact that $\max(x_1, x_2, x_3, x_4) = \max(\max(x_1, x_2), \max(x_3, x_4))$ and $\max(x_1, x_2) = \max(x_1 - x_2, 0) + \max(x_2 - \underline{x}_2, 0) + \underline{x}_2$. Therefore, we can represent the Maxpooling layer by a composition of pairwise maximum functions which can themselves be decomposed into a combination of linear and ReLU layers. The projection operations onto the convex hull of the ReLU and linear layers are closed-form as shown in Appendix C.1 and Section 2.4, respectively.

### C.8 Residual connection

We consider a typical residual connection $\phi_k(y_k, y_i) = y_k + W_i y_i + b_i$ for $i < k$, where $y_i$ is an arbitrary layer before $y_k$ and $W_i, b_i$ is any affine transformation (i.e. a 1 by 1 convolution for upsampling or downsampling layer frequently used in residual connections with differently sized feature maps). Then, we can write this as

$$\phi_k(y_{ki}) = W_{ki} y_{ki} + b_i, \tag{30}$$

where $y_{ki} = \begin{bmatrix} y_k \\ y_i \end{bmatrix}$ and $W_{ki} = [I \quad W_i]$. Then, the projection operator reduces to the affine case but with weights $W_{ki}, b_i$ for input $y_{ki}$.

### C.9 Batch normalization and average pooling

The batch normalization and average pooling layers that appear in the architecture of the residual networks, e.g., ResNet18 applied in this paper, are essentially affine mapping layers whose weights and biases can be extracted accordingly. Therefore, the projection onto the graph of these two layers has a closed-form solution as shown in Eq. (11).

## D Implementation Details

### D.1 Convergence and Stopping Criterion

The DeepSplit algorithm converges to the optimal solution of the convex problem (6) under mild conditions. Specifically, when $J$ is closed, proper and convex, and when the sets $\mathcal{S}_k$ (convex outer approximations of the graph of the layers) along with $\mathcal{X}$ are closed and convex, we can resort to the convergence results of ADMM (Boyd et al., 2011).

For the LP relaxation (6) of a feed-forward neural network, the primal and dual residuals are defined as

$$r_p = \sum_{k=0}^{\ell-1} \{\|y_k^+ - x_k^+\|^2 + \|x_{k+1}^+ - z_k^+\|^2\} \tag{31}$$

$$r_d = \rho \sum_{k=1}^{\ell-1} \|(y_k^+ - y_k) + (z_{k-1}^+ - z_{k-1})\|_2^2 + \rho \left(\|y_0^+ - y_0\|_2^2 + \|z_{\ell-1}^+ - z_{\ell-1}\|_2^2\right).$$

These are the residuals of the optimality conditions for (6) and converge to zero as the algorithm proceeds. A reasonable termination criterion is that the primal and dual residuals must be small, i.e. $r_p \leq \epsilon_p$ and $r_d \leq \epsilon_d$, where $\epsilon_p > 0$ and $\epsilon_d > 0$ are tolerance levels (Boyd et al., 2011, Chapter 3). These tolerances can be chosen using an absolute and relative criterion, such as

$$\epsilon_p = \sqrt{p}\, \epsilon_{abs} + \epsilon_{rel} \max\{(\|x_0\|_2^2 + 2\sum_{i=1}^{\ell-1} \|x_i\|_2^2 + \|x_\ell\|_2^2)^{\frac{1}{2}} + (\sum_{i=0}^{\ell-1}(\|y_i\|_2^2 + \|z_i\|_2^2)^{1/2}\}$$

$$\epsilon_d = \sqrt{n}\, \epsilon_{abs} + \epsilon_{rel}(\|\lambda_0\|_2^2 + \sum_{i=1}^{\ell-1} \|\lambda_i + \mu_{i-1}\|_2^2 + \|\mu_{\ell-1}\|_2^2)^{\frac{1}{2}},$$

where $p = n_0 + 2\sum_{i=1}^{\ell-1} n_i + n_\ell$, $n = \sum_{i=0}^{\ell} n_i$, $\epsilon_{abs} > 0$ and $\epsilon_{rel} > 0$ are absolute and relative tolerances. Here $n$ is the dimension of $\mathbf{x}$, the vector of primal variables that are updated in the first step of the algorithm, and $p$ is the total number of consensus constraints.

For neural networks with general computational graphs such as residual networks, the primal and dual residuals as well as the stopping criteria have different representations from what's shown above. But with our proposed splitting method, these representations are easy to derive from (Boyd et al., 2011, Chapter 3).

### D.2 PARAMETER SELECTION

A proper selection of the augmentation constant $\rho$ has a dramatic effect on the convergence of the method. Large values of $\rho$ enforces consensus more quickly, yielding smaller primal residuals but larger dual ones. Conversely, smaller values of $\rho$ leads to larger primal and smaller dual residuals. A commonly used heuristic to make this trade-off is residual balancing He et al. (2000), in which the penalty parameter varies adaptively based on the following rule:

$$\rho^+ = \begin{cases} \tau\rho & \text{if } r_p > \mu r_d \\ \rho/\tau & \text{if } r_d > \mu r_p \\ \rho & \text{otherwise,} \end{cases}$$

where $\mu, \tau > 1$ are parameters. In our experiments, we found this rule to be effective in speeding up the practical convergence.

### D.3 CONVOLUTIONAL LAYERS

The projection step for affine layers from (11) requires multiplication by the matrix $(I_{n_k} + W_k^\top W_k)^{-1}$ for that layer, where $n_k$ is the number of neurons in the layer. When handling networks with convolutional layers on image data, $n_k$ can easily exceed tens of thousands, so the resulting matrix and its inversion can exceed reasonable memory and computational constraints.

To make the update step practical and in line with typical computational costs of deep convolutional networks, we replace the typical deep learning convolution with an equivalent circular convolution. Specifically, let $f_{\text{conv}}$ be a typical strided convolution with padding. We can rewrite $f_{\text{conv}}$ as three sequential updates using a circular convolution as $f_{\text{conv}} = f_{\text{post}} \circ f_{\text{circ}} \circ f_{\text{pad}}$, where $f_{\text{pad}}$ is a padding step, $f_{\text{circ}}$ is a circular convolution, and $f_{\text{post}}$ is a

post-processing step that performs cropping, downsampling, and adds any bias term from $f_{\text{conv}}$.

We can now treat these three updates as separate, individual layers in our ADMM algorithm. The key observation is that we can use the convolution theorem to implement the ADMM update for the circular convolution $f_{\text{circ}}$ efficiently. Specifically, for an input of size $n_k \times n_k$, the projection step from (11) for a circular convolution can be calculated in $O(n_k^2 \log n_k)$ using the fast Fourier transform. We provide the specific details of this procedure in Appendix E.

## E    FFT IMPLEMENTATION FOR CIRCULAR CONVOLUTIONS

In order to efficiently implement convolutional layers, recall that we decompose a convolution into the following three steps:

$$f_{\text{conv}} = f_{\text{post}} \circ f_{\text{circ}} \circ f_{\text{pad}} \tag{32}$$

We now discuss in detail how to efficiently perform the $(y, z)$ update for multi-channel, circular convolutions $f_{\text{circ}}$ using Fourier transforms. We begin with the single-channel setting, and then extend our procedure to the multi-channel setting. See Appendix C for details regarding the ADMM projection step for $f_{\text{pad}}$ and $f_{\text{post}}$.

### E.1    SINGLE-CHANNEL CIRCULAR CONVOLUTIONS

Let $U$ represent the discrete Fourier transform (DFT) as a linear operator, and let $W$ be the weight matrix for the circular convolution $f_{\text{circ}}(x) = W * x$. Then, using matrix notation, the convolution theorem states that

$$f_{\text{circ}}(x) = W * x = U^*(UW \cdot Ux) = U^*DUx \tag{33}$$

where $D = \text{diag}(UW)$ is a diagonal matrix containing the Fourier transform of $W$ and $U^*$ is the conjugate transpose of $U$. Then, we can represent the inverse operator from (11) as

$$(I + f_{\text{circ}}^\top f_{\text{circ}})^{-1} = U^*(I + DD)^{-1}U \tag{34}$$

Since $(I + DD)$ is a diagonal matrix, its inverse can be computed quickly by simply inverting the diagonal elements, and requires storage space no larger than the original kernel matrix. Thus, multiplication by the inverse matrix for a circular convolution reduces to two DFTs and an element-wise product. For an input of size $n \times n$, this step has an overall complexity of $O(n^2 \log n)$ when using fast Fourier transforms.

### E.2    MULTI-CHANNEL CIRCULAR CONVOLUTIONS

We now extend the operation for single-channel circular convolutions to multi-channel, which is typically used in convolutional layers found in deep vision classifiers. Specifically, for a circular convolution with $n$ input channels and $m$ output channels, we have

$$f_{\text{circ}}(x)_j = \sum_{i=1}^{n} W_{ij} * x_i \tag{35}$$

where $f_{\text{circ}}(x)_j$ is the $j$th output channel output of the circular convolution, $W_{ij}$ is the kernel of the $i$th input channel for the $j$th output channel, and $x_i$ is the $i$th channel of the input $x$. The convolutional theorem again tells us that

$$f_{\text{circ}}(x)_j = \sum_{i=1}^{n} U^*D_{ij}Ux_i \tag{36}$$

where $D_{ij} = \text{diag}(UW_{ij})$. This can be re-written more compactly using matrices as

$$f_{\text{circ}}(x) = \bar{U}^*\bar{D}\bar{U}\bar{x} \tag{37}$$

where

- $\bar{U} = \begin{bmatrix} U & \cdots & 0 \\ \vdots & \ddots & \vdots \\ 0 & \cdots & U \end{bmatrix}$ is a block diagonal matrix with $n$ copies of $U$ along the diagonal

- $\bar{U}^* = \begin{bmatrix} U^* & \cdots & 0 \\ \vdots & \ddots & \vdots \\ 0 & \cdots & U^* \end{bmatrix}$ is a block diagonal matrix with $m$ copies of $U$ along the diagonal

- $\bar{D} = \begin{bmatrix} D_{11} & \cdots & D_{n1} \\ \vdots & \ddots & \vdots \\ D_{1m} & \cdots & D_{nm} \end{bmatrix}$ is a block matrix with diagonal blocks where the $ij$th block is $D_{ij}$

- $\bar{x} = \begin{bmatrix} x_1 \\ \vdots \\ x_n \end{bmatrix}$ is a vertical stacking of all the input channels.

Then, we can represent the inverse operator from (11) as

$$(I + f_{\mathrm{circ}}^\top f_{\mathrm{circ}})^{-1} = \bar{U}^*(I + \bar{D}\bar{D})^{-1}\bar{U} \tag{38}$$

where $I + \bar{D}\bar{D}$ is a block matrix, where each block is a diagonal matrix. The inverse can then be calculated by the inverting sub-matrices formed from sub-indexing the diagonal components. Specifically, let $\bar{D}_{j::p}$ be a slice of $\bar{D}$ containing elements spaced $m$ elements apart in both column and row directions, starting with the $j$th item. For example, $\bar{D}_{0::p}$ is the matrix obtained by taking the top-left most element along the diagonal of every block. Then, for $j = 1 \ldots m$, we have

$$(I + \bar{D}\bar{D})^{-1}_{j::p} = \left((I + \bar{D}\bar{D})_{j::p}\right)^{-1} \tag{39}$$

Thus, calculating this matrix amounts to inverting a batch of $p$ matrices of size $m \times m$. For typical convolutional networks, $m$ is typically well below $1,000$, and so this can be calculated quickly. Further note that this only needs to be calculated once as a pre-computation step, and can be reused across different inputs and objectives for the network.

**Memory and runtime requirements** In practice, we do not store the fully-expanded block diagonal matrices; instead, we omit the zero entries and directly store the the diagonal entries themselves. Consequently, for an input of size $p$, the diagonal matrices require storage of size $O(mnp)$, and the inverse matrix requires storage of size $O(m^2p)$. Since the discrete Fourier transform can be done in $O(p \log p)$ time with fast Fourier transforms, the overall runtime of the precomputation step to form the matrix inverse is the cost of the initial DFT and the batch matrix inverse, or $O(nmp \log p + m^3 p)$. Finally, the runtime of the projection step is $O((n+m)p \log p + n^2 mp)$, which is the respective costs of the DFT transformations $\bar{U}$ and $\bar{U}^*$, as well as the multiplication by $\bar{D}$. Since the number of channels in a deep network are typically much smaller than the size of the input to a convolution (i.e. $n < p$ and $m < p$), the costs of doing the cyclic convolution with Fourier transforms are in line with typical deep learning architectures.

## F    EXPERIMENTAL DETAILS

### F.1    ARCHITECTURES, DATASETS, AND TRAINING SPECIFICS

**MNIST** We consider two ReLU based architectures. The first one is a fully connected network with layer sizes $784 - 600 - 400 - 200 - 100 - 10$ which we denote MNIST-A. It is more than triple the size of that considered by (Dathathri et al., 2020) with one additional layer. It is, however, still small enough such that Gurobi is able to solve the LP relaxation, and allows us to compare our running time against Gurobi. The second one is a convolutional network

Table 3: Approximate verification time for solving the LP relaxation through ADMM. Epsilons are evenly spaced within the range.

| Model | # of epsilon | Epsilon range | Time (hrs) |
|---|---|---|---|
| MNIST-A | 10 | $[0.01, 0.1]$ | 13.12 |
| MNIST-B | 10 | $[0.01, 0.1]$ | 19.92 |
| CIFAR10 | 8 | $[0.5/255, 4/255]$ | 50.69 |
| Bankheist | 10 | $[0.1/255, 1/255]$ | 4.96 |
| Roadrunner | 10 | $[0.1/255, 1/255]$ | 52.33 |
| Pong | 10 | $[0.1/255, 1/255]$ | 31.06 |

which uses the small convolutional architecture from (Wong et al., 2018) and consists of two convolutional layers of size $16 - 32$ with kernel sizes $4 - 4$, strides $2 - 2$, and padding $1 - 1$. We denote this network by MNIST-B. This architecture is comparable to the convolutional architecture considered by (Dathathri et al., 2020), and allows us to compare our running time against the gradient-based SDP solver.

We train both models with an $\ell_\infty$ PGD adversary at radius $\epsilon = 0.1$, using 7 steps of size $\alpha = 0.02$, with batch size 100 for 100 epochs. We use the Adam optimizer with a cyclic learning rate (maximum learning rate of 0.005), and both models achieve a clean accuracy of 99%.

**CIFAR10**  For CIFAR10, we use the large convolutional architectures from (Wong et al., 2018), which consists of four convolutional layers with $32 - 32 - 64 - 64$ channels, with strides $1 - 2 - 1 - 2$, kernel sizes $3 - 4 - 3 - 4$, and padding $1 - 1 - 1 - 1$. This is followed by three linear layers of size $512 - 512 - 10$. This is significantly larger than the CIFAR10 architectures considered by (Dathathri et al., 2020), and has sufficient capacity to reach 43% adversarial accuracy against an $\ell_\infty$ PGD adversary at $\epsilon = 8/255$.

The model is trained against a PGD adversary with 7 steps of size $\alpha = 2/255$ at a maximum radius of $\epsilon = 8.8/255$, with batch size 128 for 200 epochs. We used the SGD optimizer with a cyclic learning rate (maximum learning rate of 0.23), momentum 0.9, and weight decay 0.0005. The model achieves a clean test accuracy of 71.8%.

**State-robust RL**  We use the pretrained, adversarially trained, DQNs released by (Zhang et al., 2020) from

$$\texttt{https://github.com/chenhongge/SA\_DQN}$$

which has not included any license information up to the date of submission of this paper. These models were trained to be robust at $\epsilon = 1/255$ with a PGD adversary for the Atari games Pong, Roadrunner, Freeway, and BankHeist. Each input to the DQN is of size $1 \times 84 \times 84$, which is more than double the size of CIFAR10. The DQN architectures are convolutional networks, with three convolutional layers with $32 - 64 - 64$ channels, with kernel sizes $8 - 4 - 3$, strides $4 - 2 - 1$, and no padding. This is followed by two linear layers of size $512 - K$, where $K$ is the number of discrete actions available in each game.

F.2  TIMING EXPERIMENTS

**Runtime summary**: We certify the robustness of classifiers for MNIST and CIFAR10 and DQNs in Atari games to $\ell_\infty$ perturbations over a range of radii. The specific verification setups can be found in Appendix F.3 and F.4. We report the approximate runtime for all experiments in Table 3.

**Effects of ADMM parameters**: In this subsection, we demonstrate the effects of the algorithm parameters on the convergence of ADMM through two networks: MNIST-A (fully-connected) and MNIST-B (convolutional). Specifically, we focus on the choice of $\rho$ and the application of residual balancing.

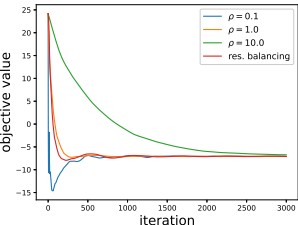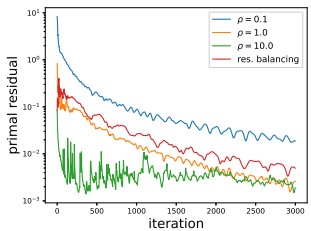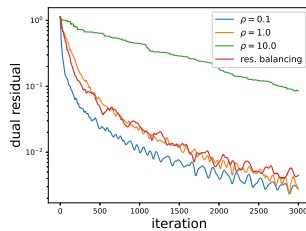

Figure 4: The objective values (left), primal residuals (middle), and dual residuals (right) of ADMM under different augmentation parameters $\rho$ on the MNIST-A (fully-connected) network.

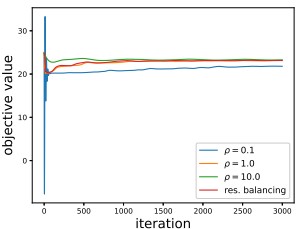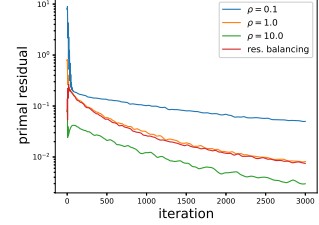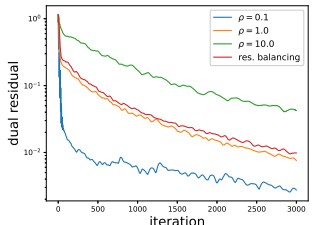

Figure 5: The objective values (left), primal residuals (middle), and dual residuals (right) of ADMM under different augmentation parameters $\rho$ on the MNIST-B (convolutional) network.

We conduct our experiment on the 1938-th example which is randomly chosen from the MNIST test data set. For this example, both MNIST-A and MNIST-B predicts its class (number 4) correctly. We add an $\ell_\infty$ perturbation of radius $\epsilon = 0.02$ to the input image and verify if the network outputs are robust with respect to class number 3. The maximum number of iterations is restricted to 3000. The objective values, primal and dual residuals of ADMM for MNIST-A and MNIST-B under different augmentation parameters $\rho$ are plotted in Figure 4 and 5, respectively. The residual balancing in this experiment is applied with initial $\rho = 10.0$, $\tau = 2$, and $\mu = 10$ as described in Section D.2.

The update rule of ADMM suggests that a large value of $\rho$ tends to produce small primal residuals since it puts a large penalty on the violation of the primal feasibility. However, the dual residuals for such $\rho$ tends to be larger. Conversely, a small $\rho$ tends to reduce the dual residuals at the cost of larger primal residuals. This phenomenon is illustrated empirically with the fully-connected network MNIST-A in Figure 4 and the convolutional network MNIST-B in Figure 5.

Since ADMM terminates when both the primal and dual residuals are small enough, in practice we prefer to choose the augmentation parameter $\rho$ not too large or too small in order to balance the reduction in the primal and dual residuals. An effective way to choose a good $\rho$ is residual balancing, which tries to keep the primal and dual residuals close during the ADMM updates by adjusting $\rho$ online. In both Figure 4 and Figure 5, ADMM with residual balancing is initialized with $\rho = 10.0$ and shows significant improvement in convergence rate compared with the case of constant $\rho = 10.0$. As observed in our other experiments, with residual balancing, ADMM becomes insensitive to the initialization of $\rho$ and usually achieves a good convergence rate.

In all of the test accuracy certification results reported in this paper, we initialize $\rho = 1.0$ and apply residual balancing when running ADMM. In different experiments, the stopping criterion parameters $\epsilon_{\mathrm{abs}}, \epsilon_{\mathrm{rel}}$ of ADMM are chosen by trial-and-error to achieve a balance between the accuracy and the running time of the algorithm. Although ADMM has convergence guarantees for its objective values, primal and dual residuals, it may take too many iterations for ADMM to achieve a solution of high accuracy (Boyd et al., 2011).

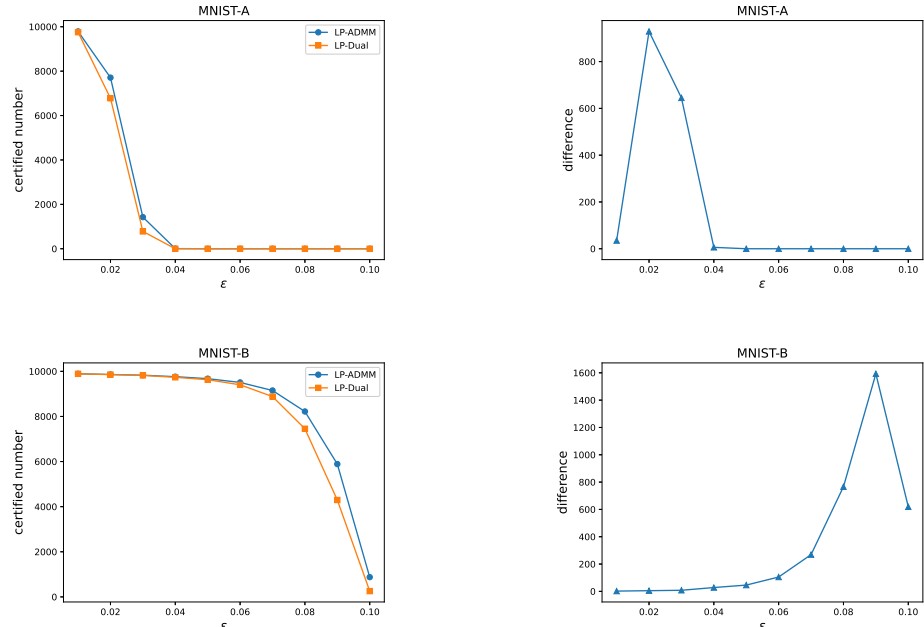

Figure 6: The numbers of certified examples (left column) of ADMM and the linear-based bounds (Wong et al., 2018; Xu et al., 2020) for the MNIST-A (upper row) and MNIST-B (lower row) networks and their differences (right column) for each $\epsilon \in \{0.01, 0.02, \cdots, 0.10\}$.

### F.3 IMAGE CLASSIFICATION

In this section, we report the details of certified test accuracy of our proposed method (ADMM) and the scalable linear-based bounds on LP relaxations (Linear) (Wong et al., 2018; Xu et al., 2020) in the image classification tasks of MNIST and CIFAR10.

**MNIST**: For both the fully-connected network MNIST-A and the convolutional network MNIST-B, we apply LP-ADMM with $\epsilon_{\mathrm{abs}} = 10^{-3}, \epsilon_{\mathrm{rel}} = 10^{-3}$ as the stopping criterion, and residual balancing with the initial $\rho = 1.0$. For each classifier network, we go through the 10000 examples in the MNIST test data set and a range of $\ell_\infty$ perturbation radii $\epsilon$ to count the number of certified examples. To make the counting more efficient, we (i) search over $\epsilon$ in a descending order since examples that are robust for a larger $\epsilon$ are also robust for a smaller $\epsilon$, and (ii) only apply ADMM on examples that cannot be verified by Linear (Wong et al., 2018; Xu et al., 2020) since Linear (Wong et al., 2018; Xu et al., 2020) gives a more relaxed bound than the LP-relaxation. For each batch of test examples, ADMM solves ten optimization problems of the form (5) with the linear objective function given with respect to each prediction class.

For a range of $\epsilon \in \{0.01, 0.02, \cdots, 0.10\}$, the numbers of verified examples of ADMM and Linear (Wong et al., 2018; Xu et al., 2020) and their differences are shown in Figure 6 for MNIST-A and MNIST-B, respectively. Exact certified accuracy is reported in Table 4 for a range of $\epsilon$.

**CIFAR10**: On the CIFAR10 data set, we compare ADMM and Linear (Wong et al., 2018; Xu et al., 2020) following the same process as on the MNIST data set. The stopping criterion for LP-ADMM is set as $\epsilon_{\mathrm{abs}} = 10^{-4}, \epsilon_{\mathrm{rel}} = 10^{-4}$ and the range of $\epsilon$ is set as $\{0.5/255, 1.0/255, \cdots, 4.0/255\}$. The numbers of certified examples of ADMM and Linear (Wong et al., 2018; Xu et al., 2020) and their differences are shown in Figure 7.

**Further details for baselines in Table 1** In Table 1, we report the certified accuracy of solving the LP exactly with ADMM in comparison to a range of baselines. We compare with methods that have previously demonstrated the ability to bound networks of this size: fast

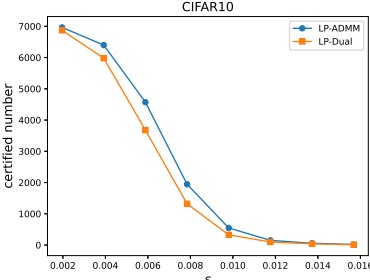 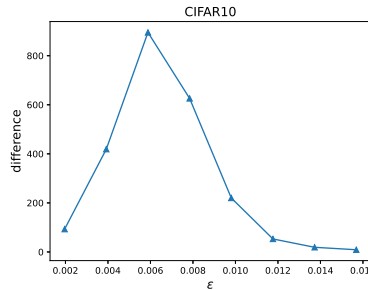

Figure 7: The number of certified examples (left) of ADMM and the linear-based bounds (Wong et al., 2018; Xu et al., 2020) for the CIFAR10 network and their differences (right) for each $\epsilon \in \{0.5/255, 1.0/255, \cdots, 4.0/255\}$.

Table 4: Certified accuracy of PGD-trained models on MNIST when using a computationally cheap bound (linear-based bounds (Wong et al., 2018; Xu et al., 2020)) vs. our method (ADMM)

| Model | Epsilon | Certified test accuracy (%) | | |
| --- | --- | --- | --- | --- |
| | | **Linear** | **ADMM** | **Diff** |
| MNIST-A | 0.02 | 67.8 | 77.1 | **9.3** |
| | 0.03 | 7.8 | 14.2 | **6.5** |
| MNIST-B | 0.07 | 88.8 | 91.5 | **2.7** |
| | 0.08 | 74.5 | 82.2 | **7.7** |
| | 0.09 | 43.0 | 58.9 | **15.9** |
| | 0.10 | 2.6 | 8.8 | **6.2** |

bounds of the LP (Linear) (Wong et al., 2018; Xu et al., 2020), and interval bounds (IBP) (Gowal et al., 2018). We additionally compare to a suite of Lagrangian-based baselines, whose effectiveness at this scale was previously unknown. These methods include supergradient ascent (Adam) (Bunel et al., 2020b), dual supergradient ascent (Dual Adam) (Dvijotham et al., 2018) and a variant thereof (Dual Decomp Adam) (Bunel et al., 2020b), and a proximal method (Prox) (Bunel et al., 2020a). Lagrangian-based baselines were given the same computational budget as the ADMM solver, with further details in Appendix G.

### F.4 REINFORCEMENT LEARNING

We compare the tightness of ADMM and Linear (Wong et al., 2018; Xu et al., 2020) by verifying the robustness of DQNs on three Atari game benchmarks: BankHeist, Roadrunner, and Pong [3]. The DQNs applied in these experiments are introduced in Appendix F.1.

For each benchmark, we collect $10,000$ frames (each with dimension $1 \times 84 \times 84$) across 100 episodes using the natural policy with 100 frames sampled randomly from each episode as the test data set. Note that the input images to the DQNs are already pre-processed such that the pixel values are normalized to $[0, 1]$ with a single channel.

In each game, for each frame in the sampled data set, we verify if the DQN does not change its actions when an $\ell_\infty$ perturbation of various radii is added to the frame which is the state observation of the agent. The number of actions in BankHeist, Roadrunner and Pong are $6, 6, 4$, respectively. Essentially, we reduce the verification of the DQN with finite discrete action space to the same setting as verifying classifiers. Therefore, we apply the same verification process as descried in Appendix F.3 with $\epsilon \in \{0.1/255, 0.2/255, \cdots, 1.0/255\}$ on

---

[3](Zhang et al., 2020) consider one additional RL setting (Freeway). However, the released PGD-trained DQN is completely unverifiable for nearly all epsilons that we considered.

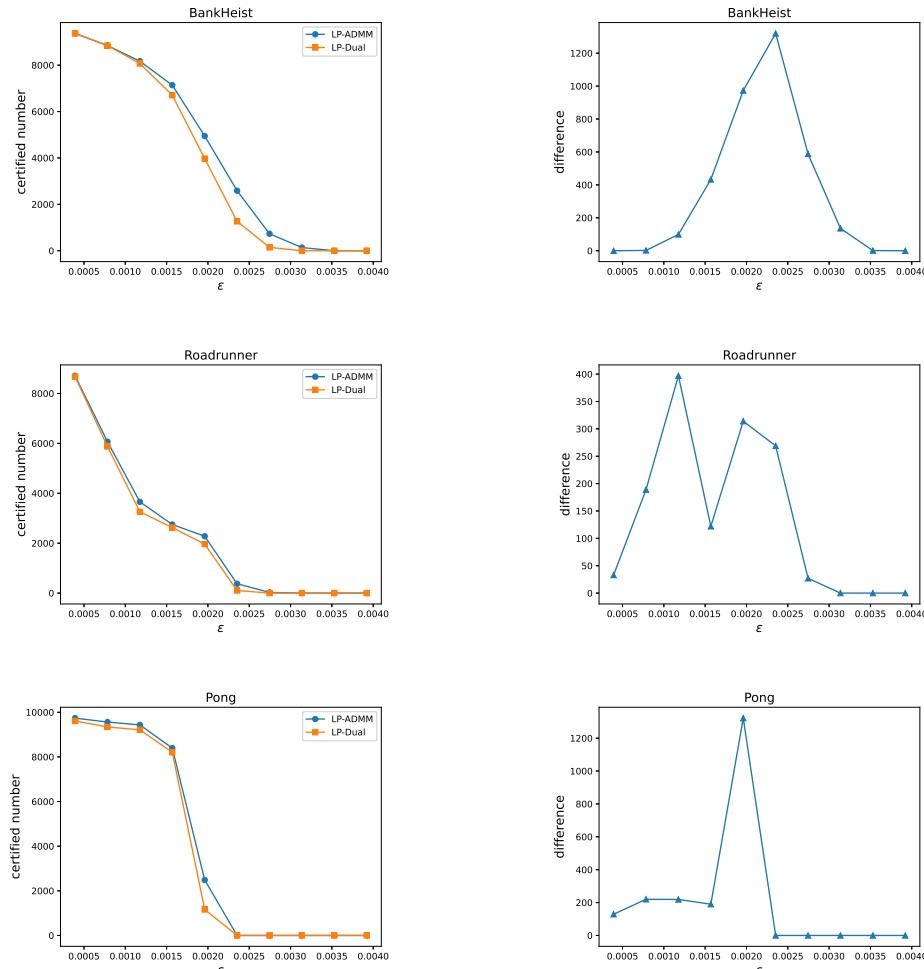

Figure 8: The numbers of certified examples (left column) of ADMM and the linear-based bounds (Wong et al., 2018; Xu et al., 2020) and their differences (right column) in the verification of DQNs from the BankHeist (first row), Roadrunner (second row) and Pong (third row) benchmarks. The range of $\epsilon$ is given by $\{0.1/255, 0.2/255, \cdots, 1.0/255\}$.

all three tasks. The numbers of certified examples of ADMM and Linear (Wong et al., 2018; Xu et al., 2020) and their differences on each task are plotted in Figure 8.

## G  ADDITIONAL DETAILS ON COMPARISON TO LAGRANGIAN-BASED METHODS

In Section 4.1, we compared the certified test accuracy of ADMM with a suite of Lagrangian-based methods (Bunel et al., 2020a) on CIFAR10. These Lagrangian-based baselines were implemented using the codes at

https://github.com/oval-group/decomposition-plnn-bounds

under the MIT license. As mentioned in Section 3.2, these baselines require solving an inner optimization problem through iterative algorithms. In the experiments of Table 1, the number of iterations of different iterative algorithms are bounded separately such that each Lagrangian-based method listed in Section 4.1 has runtime of 9 seconds to finish verifying one example.

Our ADMM solver averages 9 seconds runtime per example in this verification task, which is the same as the average runtime of the Lagrangian-based methods, with the stopping criterion of $\epsilon_{abs} = 10^{-4}, \epsilon_{rel} = 10^{-3}$ and $\rho = 1.0$.

**Architecture size.** We note that the architecture we verify in this paper is 10x larger than considered by (Bunel et al., 2020b). However, the released framework could not handle this model size due to their implementation of intermediate bounds. In order to handle this model size, we externally calculated the intermediate bounds (using (Wong et al., 2018)) and loaded these into the framework for the Lagrangian-based baselines.

## H  Speed comparison

**Comparison with LP solver**: To demonstrate the effectiveness of GPU-acceleration in the DeepSplit algorithm, we compare the runtime of DeepSplit and Gurobi in solving LP relaxations that bound the output range of MNIST-A network (the fully connected network defined in Appendix F.1) with $\ell_\infty$ perturbations in the input. Specifically, for a given example in the MNIST test data set, we apply DeepSplit/Gurobi layer-by-layer to find the tightest pre-activation bounds under the LP-relaxation.

Recall that the MNIST-A network is a fully-connected netowrk with architecture $784 - 600 - 400 - 200 - 100 - 10$. With the Gurobi solver, we need to solve $2 \times 600$ LPs sequentially to obtain the lower and upper bounds for the first activation layer, $2 \times 400$ LPs for the second activation layer, and so forth. With DeepSplit, the pre-activation bounds can be computed in batch and allows GPU-acceleration.

In our experiment, we fix the radius of the $\ell_\infty$ perturbation at the input image as $\epsilon = 0.02$. For the Gurobi solver, we randomly choose 10 samples from the test data set and compute the pre-activation bounds layer-by-layer. The LP relaxation is formulated in CVXPY and solved by Gurobi v9.1 on an Intel Core i7-6700K CPU, which has 4 cores and 8 threads. For each example, the total solver time of Gurobi is recorded with the average solver time being 275.9 seconds. For the DeepSplit method, we compute the pre-activation bounds layer-by-layer on 19 randomly chosen examples. The algorithm applies residual balancing with the initial $\rho = 1.0$ and the stopping criterion is given by $\epsilon_{abs} = 10^{-4}, \epsilon_{rel} = 10^{-3}$. The total running time of DeepSplit is 717.9 seconds, with 37.8 seconds per example on average. With the GPU-acceleration, out method achieves 7x speedup in verifying NN properties compared with the commercial-grade Gurobi solver.

**Comparison with SDP solver**: Our method is significantly faster than solving the convex relaxation through other approaches. For example, it took our method approximately 20 hours to verify all MNIST test set images for the convolutional architecture at 10 different epsilon values (See Table 3), taking on average 2 hours per epsilon to verify all 10,000 examples. See F.2 for the running time summary of each experiment. In contrast, the SDP relaxation (Raghunathan et al., 2018) takes 3 hours to verify 500 examples at one epsilon value for a similarly sized CNN. This highlights the difference in speed and scalability of solving the LP relaxation over the SDP one, albeit at the cost of looser verification guarantees.

**Comparison with branch-and-bound solver**: Branch-and-bound methods have exponential runtime as opposed to the polynomial runtime of LP-solvers. In practice, this leads to significantly longer solve times that has largely limited these methods to the small network regime. For example, the fastest branch-and-bound method from Bunel et al. (2020a) takes on average 6 minutes per example to verify a small convolutional network, whereas our solver takes on average 9 seconds per example (40x faster) to verify an order-of-magnitude larger network.

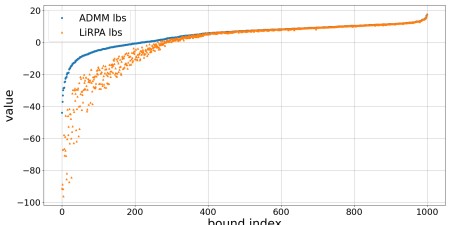 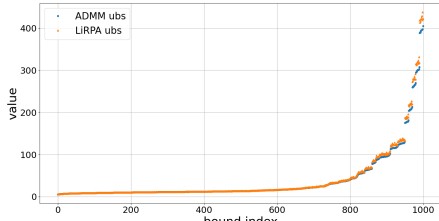

Figure 9: ResNet18 output lower (left) and upper (right) bounds obtained using either ADMM (blue dots) and LiRPA (orange triangles) from the first 100 CIFAR10 test examples. Higher lower bounds and lower upper bounds are better. For ease of visualization, bounds are sorted in ascending order according to the ADMM bound.

Table 5: Comparison of computed bounds using LiRPA (Xu et al., 2020)) vs. our method (ADMM) on a ResNet18 for CIFAR10.

|  | **Linear** | **ADMM** | Diff |
|---|---|---|---|
| Upper bounds | $39.87 \pm 66.42$ | $38.02 \pm 62.21$ | $2.32\% \pm 2.35\%$ |
| Lower bounds | $0.04 \pm 17.02$ | $4.76 \pm 7.56$ | $31.61\% \pm 43.38\%$ |

## I  SCALABILITY

To highlight the scalability of our approach, we consider a ResNet18 network trained on CIFAR10 whose max pooling layer is replaced by a down-sampling convolutional layer for comparison with LiRPA (Xu et al., 2020) [4] (codes available at

https://github.com/KaidiXu/auto_LiRPA

under the BSD 3-Clause "New" or "Revised" license), which is capable of computing provable linear bounds for the outputs of general neural networks and is the only method available so far that can handle ResNet18. The ResNet18 is adversarially trained using the fast adversarial training code from (Wong et al., 2020).

In our experiments, for the first 100 test examples in CIFAR10, we use LiRPA to compute the preactivation bounds for each ReLU layer in ResNet18 and then apply ADMM to compute the lower and upper bounds of ResNet18 outputs (there are 10 outputs corresponding to the 10 classes of the dataset). The ADMM is run with stopping criterion $\epsilon_{abs} = 10^{-5}, \epsilon_{rel} = 10^{-4}$ and augmentation parameter $\rho = 1.0$. With $\epsilon = 1/255$, the lower/upper bounds of ADMM from these 100 examples are arranged in an ascending order in Figure 9 and are compared with those obtained by LiRPA. We observe that the bounds from ADMM are consistently tighter than those from LiRPA. The average running time of ADMM across the 100 examples is 2312 seconds. Overall bound statistics are shown in Table 5.

---

[4]The max pooling layer has not been considered in the implementation of LiRPA by the submission of this paper.

