# OpenReview forum: "DeepSplit: Scalable Verification of Deep Neural Networks via Operator Splitting"
_ICLR.cc/2022/Conference — ICLR 2022 Submitted_

### Official Review · Reviewer_aJXo · 2021-10-31

**Correctness:** 2
**Technical Novelty And Significance:** 2
**Empirical Novelty And Significance:** 2
**Recommendation:** 3
**Confidence:** 5

**Main Review:**

---- Pros ----

As implied in section 3, DeepSplit is an improvement on two closely-related dual solvers for the same relaxation (the convex hull of element-wise activation functions):
- (Dvijotham et al. 2018) presented a dual solver based on the Lagrangian relaxation of problem (3), which converges to the same bounds as DeepSplit, with a rate of O(1/sqrt(T)).
- (Bunel et al. 2020a) first employed variable splitting and an Augmented Lagrangian on the same relaxation, improving upon the dual solver from (Dvijotham et al. 2018). While the Augmented Lagrangian is differentiable, the inner problems require the use of an iterative optimisation algorithm.
- DeepSplit adds a second variable split to the formulation from (Bunel et al. 2020a) so that the inner problems enjoy a closed-form solution. This results in faster empirical convergence, and attains the O(1/T) rate.

Suggestion: perhaps inverting the ordering of sections 3 and 2 could improve the paper's readability.

---- Cons ----

The paper repeatedly claims that DeepSplit solves the considered relaxation *exactly*, seemingly contrasting this with other approaches. However, the methods from (Dvijotham et al. 2018) and (Bunel et al. 2020a) enjoy the same property, the only difference being a slower empirical convergence rate. As a consequence, I believe the authors should tone down claims such as that they compute bounds for "networks whose convex relaxations were previously impossible to solve exactly due to their size".

Furthermore, in neural network verification, one is typically concerned with verifying the largest number of properties in the smallest possible time, rather than solving a given relaxation exactly. Suboptimal yet inexpensive bounds are then potentially preferable to solving a given relaxation to optimality, especially considering that running a fixed small number of branch and bound steps will significantly tighten the bounds (see for instance (Wang et al. 2021)). In other words, methods based on branch and bound can be employed as incomplete verifiers, if they are terminated early. The cost of running recent branch-and-bound methods to termination is nevertheless significantly smaller than what the authors claim: BaDNB from (De Palma et al. 2021) verifies a large number of properties in under 10s for a network of 50k activations. Similar (or better) results were obtained by the participants of VNN COMP 2020, or VNN COMP 2021. I believe this is in the same order of magnitude as the network size employed for Table 1, for which the reported runtime of DeepSplit (in the appendix) is around 10s as well.

In addition, recent works such as (Xu et al. 2021), and [Bunel et al. non-convex 2020] have shown that non-convex reformulations (lacking convergence guarantees to the exact solution of the relaxation, for ReLUs) obtain significantly better speed-accuracy trade-offs (on the same relaxation) than (Dvijotham et al. 2018) and (Bunel et al. 2020a). These newer works seemingly scale quite well with network sizes: I hence doubt that the only applicable method to a ResNet18 is non-optimized LiRPA, as claimed by the authors.

I am willing to significantly increase my score if the authors address the following points:
- Table 1 should report the average runtime in addition to the certified test accuracy. Preferably, different speed-accuracy trade-offs should be reported for iterative algorithms.
- The certified accuracy for "Dual Decomp Adam" should not be lower than the one for "Linear", as the latter can be used an initializer to the dual problem in (Bunel et al. 2020a). Are the authors taking this into account? Could DeepSplit enjoy the same initialization?
- Compare against the more recent bounding algorithms from [Bunel et al. non-convex 2020] and (Xu et al. 2021). While introduced in the context of branch and bound, the "alpha-CROWN" method, or LiRPA with optimized slopes, is a valid incomplete verifier, and the global optimum of its optimization problem coincides with DeepSplit's. Furthermore, it might yield tigther bounds than DeepSplit in case of joint optimization over intermediate bounds. The code of both algorithms is available online.
- Compare speed-accuracy trade-offs against works operating on tighter relaxations, such as [Active Set].
- Considering that the memory footprint of alpha-CROWN with fixed intermediate bounds is only marginally larger than the one for LiRPA, the authors should add this to the ResNet18 experiments.
- Complete verification performance is tightly linked to the quality of the speed-accuracy trade-offs of an incomplete verifier. It would be quite informative if the authors could provide complete verification experiments on the model from Table 1.

---- Minor comments ----
- End of page 1 "these relaxations must typically be further relaxed": this statement does not apply to (Dvijotham et al. 2018) and (Bunel et al. 2020a)
- section 1.1, typo in reference, "de2"
- Sec 3.2 "When stopping early, the primal minimization must be solved to convergence in order to compute the dual function and produce a valid bound.": this is incorrect, as a valid bound can be obtained in closed-form by minimising the Lagrangian, rather than Augmented Lagrangian.

References:
[Bunel et al. non-convex 2020] - An efficient nonconvex reformulation of stagewise convex optimization problems, NeurIPS 2020
[Active Set] - Scaling the Convex Barrier with Active Sets, ICLR 2021

**Summary Of The Paper:**

The authors present DeepSplit, a novel solver for a popular neural network convex relaxation. Relying on ADMM and on a careful problem re-formulation, the authors achieve both a O(1/T) converge rate and closed-form solutions for the inner problems, differently from previously presented solvers for the same relaxation. Computational results are presented for incomplete neural network verification, showing that DeeSplit scales to a ResNet18 and achieves better bounds in the same time compared to relevant baselines.

**Summary Of The Review:**

DeepSplit is an improvement upon two closely-related solvers for the same convex relaxation: (Dvijotham et al. 2018) and (Bunel et al. 2020a).  The work focuses on solving the relaxation exactly. However, as shown in a variety of recent papers (Xu et al. 2021), [Active Set],  [Bunel et al. non-convex 2020], (Wang et al. 2021), heuristics and switching to tighter relaxations (possibly considering branching) might verify more properties in the same time (both in incomplete and complete verification). These developments have been somewhat ignored by the authors.

The authors should put their work in perspective with more recent and scalable optimizers for the same relaxation: (Xu et al. 2021), [Bunel et al. non-convex 2020]; and with works on tighter relaxations [Active Set]. Furthermore, as commonly done for solvers of the considered relaxation, complete verification experiments are needed to fully assess the quality of DeepSplit's speed-accuracy trade-offs.

---

> ### Author Response · Authors · 2021-11-23
> **Response to Reviewer aJXo**
>
> We thank the reviewer for all the suggestions. We agree that we need to revise this sentence to properly acknowledge the contributions of previous papers. Currently available BaB methods such as (Wang et al. 2021) already consider CNN as large as the one used in our paper. However, they haven’t shown their applicability in ResNet18 which is a challenge since from our experiments the bounds provided by LiRPA are usually too coarse to use on ResNet18.
>
> We address your main questions in the review below.
>
> *1. Runtime report*:
> The average runtime of the Lagrangian methods and ADMM in Table 1 is between 9.3 s and 9.7 s per example. When the average runtime of the Lagrangian methods is controlled around 4.5 s by decreasing the number of iterations in the inner optimization, the certified robust accuracy of each Lagrangian method decreases by around 2%. We will label the average runtime in Table 1 for each method and also provide the information of the speed-accuracy trade-offs of the Lagrangian methods in our updated version of the paper.
>
> *2. Certified accuracy problem*:
> Thanks for pointing this out. We have double-checked our codes and found that there is a bug in our adaptation of the codes provided by (Bunel et al. 2020a) where we tried to manually provide intermediate bounds to the algorithm. After correction, the certified accuracies of the Lagrangian methods are increased. Those of Dual Decomp Adam are not consistently greater than those of Linear. However, the updated certified accuracies of the Lagrangian methods are still smaller than those of ADMM and our conclusion drawn from Table 1 remains unchanged. We will update Table 1 in the revised version.
>
> *3. Compare with complete verifier*:
> We have compared ADMM with the state-of-the-art BaB method alpha-beta-CROWN (Xu et al., 2021, Wang et al. 2021) which uses Fast-and-complete as a bounding subroutine in BaB in verifying the robustness of the CNN used in Table 1. The verification is done on the first 200 examples in the CIFAR10 test dataset with epsilon = 2/255.  In this experiment, ADMM uses the intermediate bounds obtained from KW and the algorithm parameters are given in Section F.3. Alpha-beta-CROWN is run with the template configuration given in https://github.com/huanzhang12/alpha-beta-CROWN with the batch size changed to 10 such that alpha-beta-CROWN does not exceed the 8GB memory limit of GPU. It turns out that ADMM takes 876 s in total and verifies 32 robust examples, while alpha-beta-CROWN takes 7263 s and verifies 77 robust examples. The gap in the number of verified examples is expected since alpha-beta-CROWN uses BaB (beta-CROWN, Wang et al. 2021) plus optimized LiRPA bounds (alpha-CROWN, Xu et al., 2021) which can reduce the conservatism from solving a single LP relaxation. However, alpha-beta-CROWN takes 8x more time to finish the verification. We will include more detailed comparison results in the revised version of the paper.
>
> *4. Compare with alpha-CROWN*:
> We have done an experiment to compare ADMM and optimized LiRPA in verifying the robustness of the CNN used in Table 1 over the first 20 examples from the CIFAR10 test dataset. The perturbation size is chosen as epsilon = 2/255. In this experiment, ADMM uses the intermediate bounds computed by the fast linear bounds from (Wong et al. 2017) to formulate the LP relaxation and verifies 4 robust examples in 423 seconds on a single GPU (GeForce GTX 1080). We then apply optimized LiRPA which updates the intermediate bounds with a learning rate 0.1 in 10 iterations and optimized LiRPA verifies 8 robust examples. Unfortunately, due to the size of the CNN, we could not run optimized LiRPA on the GPU which only has 8GB memory; instead, we ran optimized LiRPA on CPU (Intel Core i7-6700K) which gave 3622 seconds of runtime in total.
>
> *5. Active sets method*:
> Unfortunately, we don’t have enough time to run this experiment, but we want to point out that we can combine the linear layer and the ReLU activation layer and project the variables onto the convex hull of their composition in ADMM. This is exactly the design idea to improve the single neuron LP relaxation as proposed in [Active sets].

---

> > ### Comment · Reviewer_aJXo · 2021-11-29
> > **Insufficient author response**
> >
> > I thank the authors for their response.
> > However, unfortunately, I think their reply is insufficient. I will hence keep my score of 3.
> > The authors were given enough time to address the comments by running experiments and updating the draft. They instead left the draft unchanged, and provided inadequate experimental evidence via text. More detailed feedback follows:
> >
> > 3) The fact that the authors were able to run BaB-based alpha-beta-CROWN on the network from Table 1 clearly contradicts their repeated claim that BaB is not applicable for such network sizes. Furthermore, as stated in my initial reviews, alpha-beta-CROWN can be terminated early and employed as an incomplete verifier. In fact, in this case, it should have been early stopped on ADMM's average runtime. I therefore do not think the provided results are particularly informative.
> >
> > 4) Alpha-beta-CROWN coincides with optimized LiRPA, if the algorithm is modified to return at the root of the branch and bound tree. It is therefore quite surprising that the former fits into GPU memory and the latter does not.
> > Furthermore, if optimized LiRPA with the joint intermediate bounds optimization did not fit into GPU memory,  the authors should have kept the intermediate bounds fixed as stated in my initial review. The memory footprint will be comparable (if not inferior) to the one from ADMM. I hence do not believe these CPU results to be particularly informative either.
> >
> > 5) Due to the exponential size of the convex hull of the composition, the fact that ADMM could theoretically work on it does not imply it would have good speed-accuracy trade-offs. Differently from ADMM, Active Set was specifically designed to exploit the structure of the relaxation.

---

### Official Review · Reviewer_Km1h · 2021-11-02

**Correctness:** 3
**Technical Novelty And Significance:** 2
**Empirical Novelty And Significance:** 2
**Recommendation:** 5
**Confidence:** 4

**Details Of Ethics Concerns:**

No ethics concerns

**Main Review:**

This paper proposes an operator splitting method (i.e. ADMM) to deal with neural network verification problems. This is because the ADMM is scalable to large datasets. Specifically, they solve the convex relaxations of problems analytically. Extensive experiments on network verifications demonstrate outstanding performance as well as excellent scalability.  This paper has several limitations, which are discussed as follows:
1.	Some statements in the paper are unclear. For example, in the introduction, the authors state that neural networks lack formal guarantees. However, the meaning of formal guarantees is unclear, and how does it relate to safety-critical applications should be explained better. As another example,  the background of the neural network verification problem is missing. It is unclear what are potential applications of this problem, and examples should be provided to better understand required properties.
2.	The novelty of this paper should be justified better. The authors just utilize an existing optimization framework ADMM to relax nonconvex problems into convex ones, which lead to analytic solutions. As they mentioned, several papers consider similar ideas, but the difference seems subtle. it is unclear how ADMM resolves the incomplete problem (i.e. methods are guaranteed to detect all false negatives but also produce false positives). Moreover, the ADMM usually converges slowly to the solution, how do authors address this issue?
3.	Experimental details are missing.  Even though the authors show many experimental results,  the background of problems in the experiments is missing, and little information such as architectures and hyperparameter settings is provided.  So it is difficult to reproducible experimental results.


**Summary Of The Paper:**

This paper proposes ADMM for neural network verification problems. Experiments are demonstrated to show its effectiveness and scalability.

**Summary Of The Review:**

 This paper addresses an important problem. However, the unclear presentation, the lack of novelty justification, and missing experimental details should be improved to make it an acceptable paper in ICLR.

---

> ### Author Response · Authors · 2021-11-23
> **Response to Reviewer Km1h**
>
> We thank the reviewer for pointing out several aspects of the paper that needs improvement. In this paper, we want to verify that a given neural network does not change its prediction result if a perturbation is added to its input. We will revise the introduction section according to the suggestions to provide a more complete background on neural network verification.
>
> As to the second point, we aim to propose specialized algorithms to verify certain properties of a neural network, and our design is based on ADMM. How to choose the variables and do the splitting, however, is non-trivial and requires a lot of effort to validate its effectiveness. We agree that the differences between our method and other Lagrangian-based methods are subtle in that the benefits of applying ADMM are hard to notice. This issue is addressed in Section 3 where we point out that ADMM enjoys preferable theoretical convergence guarantees and has closed-form updates in each iteration.
>
> The incompleteness of the verification comes from the fact that we over-approximate the output of the neural network to verify its properties by solving a convex linear program relaxation. By the convergence guarantees, ADMM can exactly solve this LP relaxation and therefore it gives us an incomplete verifier.
>
> The convergence rate of ADMM has been shown applicable in neural network verification through several experiments in the paper. In addition to residual balancing as mentioned in the paper, there exist other ways to accelerate the convergence of ADMM in practice such as over relaxation, adaptive penalty parameter, etc. We leave as future work a further investigation of applying these methods to improve the convergence rates of ADMM in deep learning-sized problems.
>
> For the third point, the details of the experiments are given in Section F to Section I in the appendix. You can find the neural network architectures and the hyperparameters for the ADMM algorithms of each experiment there.

---

### Official Review · Reviewer_JVyx · 2021-11-04

**Correctness:** 4
**Technical Novelty And Significance:** 4
**Empirical Novelty And Significance:** 3
**Recommendation:** 8
**Confidence:** 5

**Main Review:**

# Strengths
* The paper is very clearly written, with a clear narrative explaining how the algorithm is derived (Equation 3 to 13) with a helpful summary in Algorithm 1 on how things tie together. The appendix is very detailed and provide clear explanations for how to implement each component of the algorithm efficiently.
* The contribution is clearly framed with regards to previous work. The Introduction and Related work are a good summary of the current state of neural network verification, and Section 3 does a great job at highlighting the differences between the proposed methods and the most closely related published work, highlighting why this results in significant improvements.
* The significance of the contribution is also quite important. The authors give convergence guarantees and show good convergence empirically with the optimization curves in the appendix, which previous methods could not offer. They even show that this results in improved certified accuracy (for a given time budget),
* The evaluation is done quite rigorously, on several domains (Vision / RL) and at different scales (comparison to Gurobi on small models, evaluation against related methods on Cifar10 network of a standard size, and evaluation of scalability against fast linear bound on ResNet models).

# Weaknesses
A potential improvement that this paper could benefit from is a comparison to methods such as the OptimizedLirpa bounds from the Fast and Complete paper, which the authors cite. This would not need to include comparison to the branch and bound aspect of it, but simply to the bound computation, which rely on fast linear bounds, but iteratively optimizes the slope of the relaxation. This is equivalent to solving the same problem and has been empirically shown to be quite efficient.
A comparison of the tightness / speed tradeoff that would result would be extremely useful. (The method described in this paper are valuable even if they are slower, as they provide convergence guarantees, which OptimizedLirpa bounds do not.)

**Summary Of The Paper:**

This paper proposes an efficient solver for computing the convex relaxation of Neural Networks, which is an important component of verification methods. The method is based on ADMM, with the authors proposing a novel decomposition of the optimization problem, which allows them to get closed form solutions of all the intermediate steps of the algorithm, The proposed method also improves on the guaranteed convergence rate and is shown to scale to larger networks than the ones considered in other works.

Evaluation is performed on robustness verification on Cifar10 and on Reinforcement learning models.

**Summary Of The Review:**

Very well written paper, with interesting and significant contributions, and which provide a great summary of the state of the field.

---

> ### Author Response · Authors · 2021-11-23
> **Response to Reviewer JVyx**
>
> Thank you for your time and your positive comments! We have done an experiment to compare ADMM and optimized LiRPA in verifying the robustness of the CNN used in Table 1 over the first 20 examples from the CIFAR10 test dataset. The perturbation size is chosen as epsilon = 2/255. In this experiment, ADMM uses the intermediate bounds computed by the fast linear bounds from (Wong et al. 2017) to formulate the LP relaxation and verifies 4 robust examples in 423 seconds on a single GPU (GeForce GTX 1080). We then apply optimized LiRPA which updates the intermediate bounds with a learning rate 0.1 in 10 iterations and optimized LiRPA verifies 8 robust examples. Unfortunately, due to the size of the CNN, we could not run optimized LiRPA on the GPU which only has 8GB memory; instead, we ran optimized LiRPA on CPU (Intel Core i7-6700K) which gave 3622 seconds of runtime in total. We will include a more detailed comparison between ADMM and optimized LiRPA in the revised version of the paper.

---

> > ### Comment · Reviewer_JVyx · 2021-11-29
> > **Final thoughts on the paper**
> >
> > I have read the reviews of the other reviewers, as well as the rebuttals by the author.
> >
> > I think that reviewer aJXo has a good summary of where this paper stands: It's an improvement on related solvers of the same relaxation, but from the point of the view of the task being solved (verification), it is not an impactful development. I personally estimate that this is a valuable contribution to have a solver with good convergence guarantees for network bound computation and that empirically manages to converge, and I would like this paper to be accepted, but I can see the point of the other reviewers that it has limited impact.
> >
> > If you see it as a paper in the category "optimization solvers for the per-neuron convex relaxation of neural network", this paper is quite interesting and has good results.
> > If you see it as a paper in the category "verification algorithm of robustness to adversarial examples", this paper has a limited impact compared to developments like multi-neuron relaxation and branching.

---

### Official Review · Reviewer_LjMA · 2021-11-04

**Correctness:** 3
**Technical Novelty And Significance:** 2
**Empirical Novelty And Significance:** 2
**Recommendation:** 3
**Confidence:** 4

**Main Review:**

Strengths
1. The method uses variable splitting and Lagrangian relaxation which have been used in previous verification works. However, the proposed variable splitting is novel as it allows closed-form updates for the primal variables. Thus enjoying faster convergence than Bunel et al. 2020a.

2. The authors have done a good job at discussing the differences with previous related works like Dvijotham et al. and Bunel et al. in section 3.

Concerns
1. Incomplete verification experiments not thorough:
* Are the different methods using the same intermediate bounds? I found in the supplementary that you computed intermediate bounds for Bunel et al, using KW. How are intermediate bounds computed for your method?
* The table and details are not very clear. Do the different rows correspond to the same network tested with different epsilon values?

2. Missing baselines in incomplete verification experiments:
* The paper is using weaker baselines. The authors should compare with [A] (which is a published work at ICLR21). They use a tighter relaxation (Tjandraatmadja et al., 2020) and solve it using a Lagrangian relaxation and active set based approach. Furthermore, they are outperforming solvers from Bunel et al. 2020 which has been used as a baseline in this paper.
* This also brings into question the relevance of this work. Because [A] outperforms Bunel et al. 2020. Can your method be extended to the relaxation from Tjandraatmadja et al., 2020? It is not clear that this work can outperform [A] or work with tighter relaxations.
* The authors should also compare with Fast-and-complete (Xu et al., 2021), Beta-CROWN(Wang et al. 2021) as they use the same LP relaxation as this paper, and are very effective. This should be done in both Table 1 and the scalability section, as these methods can scale to such networks.

3. Missing complete verification experiments:
* Complete verification experiments are missing. This has been done in works that have been used by the authors as a baseline (Bunel et al. 2020). Even the recent propagation-based solvers provide results on complete verification (see Xu et al., 2021 and Wang et al. 2021).
* They are quite important to judge the practical relevance of the algorithm. Branch-and-bound takes memory, speed and accuracy all into account, thus is crucial to see the relevance of the work.
* See VNN-comp [B] for recent developments in this space. The authors should use a branch-and-bound framework from the open-sourced codes from VNN-comp and plug DeepSplit into them.
* You should also add these b&b baselines in the speed-comparison experiment. b&b solvers can be used as incomplete verifiers, and that should be possible here since the network size is small.

4. Scalability experiments representation:
* Can you provide verified accuracy here? This will help in checking if the method can in fact verify this size networks or if the bounds are too large to be relevant.

5. Overclaiming
* 'focus on networks whose convex relaxations were previously impossible to solve exactly due to their size.' in Introduction seems to be an overclaim, since LirPA seems to be not doing that bad. And LirPA is not even the state-of-the-art baseline. Is this statement corresponding to experiments in Table 1 or the scalability experiment?

I am willing to update my rating depending on the authors' response to these concerns.

[A] Scaling the Convex Barrier with Active Sets Alessandro De Palma, Harkirat Behl, Rudy R Bunel, Philip Torr, M. Pawan Kumar, ICLR 2021

[B] https://sites.google.com/view/vnn20/vnncomp

**Summary Of The Paper:**

The paper proposes a novel method for the neural network verification problem. The authors use variable splitting and Lagrangian relaxation to relax the verification problem, which is then solved with ADMM. The method is GPU friendly (similar to previous work like Dvijotham et al. 2018 and Bunel et al. 2020a), and claims to solve the relaxation to optimality.

**Summary Of The Review:**

The authors have proposed a novel method for the neural network verification problem. It is better than Bunel et al 2020. However, authors have not compared against recent state-of-the-art published baselines. Also, complete verification experiments are missing.

---

> ### Author Response · Authors · 2021-11-23
> **Response to Reviewer LjMA Part 1/2**
>
> We thank the reviewer for the comments, and we are willing to address your concerns below.
>
> *Concern 1*:
> We applied the same intermediate bounds obtained by KW for ADMM and the methods from Bunel et al. in Table 1. In Table 1, the same neural network model is used for verification and different rows just correspond to different values of epsilon. We are sorry for the ambiguity in introducing the experimental setup and we will clarify this in the revised version.
>
> *Concern 2*:
> Our proposed method in the paper is highly modularized and easy to implement. However, it is not the only way to apply ADMM for neural network verification and some of its variants may lead to tighter relaxations or faster convergence. For example, we can combine the linear layer and the ReLU activation layer and project onto the convex hull of their composition in ADMM. This is exactly the design idea to improve the single neuron LP relaxation in [A]. In this paper, we choose the standard single neuron LP relaxation to demonstrate the scalability and applicability of ADMM in neural network verification. Developing ADMM and its variants for faster and tighter verification will be one of our future research directions.
>
> We have compared ADMM with the state-of-the-art BaB method alpha-beta-CROWN (Xu et al., 2021, Wang et al. 2021) which uses Fast-and-complete as a bounding subroutine in BaB in verifying the robustness of the CNN used in Table 1. The verification is done on the first 200 examples in the CIFAR10 test dataset with epsilon = 2/255.  In this experiment, ADMM uses the intermediate bounds obtained from KW and the algorithm parameters are given in Section F.3. Alpha-beta-CROWN is run with the template configuration given in https://github.com/huanzhang12/alpha-beta-CROWN with the batch size changed to 10 such that alpha-beta-CROWN does not exceed the 8GB memory limit of GPU. It turns out that ADMM takes 876 s in total and verifies 32 robust examples, while alpha-beta-CROWN takes 7263 s and verifies 77 robust examples. The gap in the number of verified examples is expected since alpha-beta-CROWN uses BaB (beta-CROWN, Wang et al. 2021) plus optimized LiRPA bounds (alpha-CROWN, Xu et al., 2021) which can reduce the conservatism from solving a single LP relaxation. However, alpha-beta-CROWN takes 8x more time to finish the verification. We will include more detailed comparison results in the revised version of the paper.
>
> As shown in the paper, ADMM enjoys preferable theoretical convergence guarantees and empirically good convergence in solving the LP relaxation, and scales to large neural networks such as Resnet18. With the fast and tighter intermediate bounds available thanks to optimized LiRPA (alpha-CROWN), we can formulate less conservative LP relaxation and apply ADMM as the solver to achieve even tighter verification results than the optimized LiRPA. ADMM can be potentially used as a bounding subroutine in the BaB algorithm for complete verification as suggested by the reviewer. We want to stress that studying ADMM in the LP relaxation is useful in demonstrating its potential in neural network verification and we will explore more of its power in future works.
>
> *Concern 3*:
> Complete verification through methods such as mixed-integer programming has worst-case exponential complexity and usually can only be applied to small networks. In our paper, we consider CNN (in Table 1) that is an order of magnitude larger than those used in (Bunel et al. 2020, Xu et al., 2021a) and the complete verification baselines in (Wang et al. 2021). Therefore, providing complete verification of the considered CNN is very time-consuming if not intractable. We agree with the reviewer that plugging ADMM as a subroutine into BaB is a very interesting and potentially powerful verification method. However, to fully demonstrate its power we still need to tune the algorithm parameters of ADMM which takes non-trivial effort. We will leave it for future research and add the discussion of the BaB baseline in our experiments in our updated draft.
>
> [A] Scaling the Convex Barrier with Active Sets Alessandro De Palma, Harkirat Behl, Rudy R Bunel, Philip Torr, M. Pawan Kumar, ICLR 2021

---

> > ### Author Response · Authors · 2021-11-23
> > **Response to Reviewer  LjMA Part 2/2**
> >
> > *Concern 4*:
> > Unfortunately, due to the size of ResNet18, the intermediate bounds obtained by LiRPA are too coarse to do robust accuracy verification. Verifying the robust accuracy of ResNet18 is very challenging and has not been done by any method as far as the authors know. To verify the robust accuracy, we need to obtain good enough intermediate bounds to formulate the LP relaxation first and then apply ADMM to exactly solve the LP relaxation. The scalability experiments provided in the paper are meant to show that ADMM can scale to solve LP relaxations of neural networks as large as ResNet18. On the other hand, the scalability of ADMM indicates that we can approach accuracy verification of ResNet18 by applying ADMM to compute the intermediate bounds in parallel across the layers which will give us the tightest single neuron LP relaxation. We leave this for future research.
> >
> > *Concern 5*:
> > We agree that this sentence needs revision since by ‘convex relaxations’ we mean the single neuron LP relaxation and (Dvijotham et al. 2018) and (Bunel et al. 2020a) have proposed methods that can solve the LP relaxation but with slower empirical convergence.

---

> > > ### Comment · Reviewer_LjMA · 2021-11-29
> > > **Reply to authors comments**
> > >
> > > I would like to thank the authors for their response. After going through their response and the other reviews, I am retaining my score of 3.
> > >
> > > While the authors were able to address some of my concerns (1 and 5), my other concerns have not been satisfactorily resolved (2,3 and 4). To summarise:
> > >
> > > Main claim: capacity to handle large networks. However, this claim is not thorough:
> > > 1. Propagation-based solvers: Authors have not shown the comparison with state-of-the-art methods like optimised LiRPA on large networks, even though I asked for it: 'This should be done in both Table 1 and the scalability section, as these methods can scale to such networks.'. Even the comparison for table 1 is not clear as they are not run for the same amount of time. Please provide these results.
> > > 2. Since the Resnet18 is not really verified, I am not sure about the relevance of the technique. Further, the authors have not toned down the overclaims in the paper despite the suggestions from the reviewers. I do not want to discourage the authors. I can see the importance of doing verification on large networks, but the authors need to spend more effort on thoroughly justifying the claim.
> > >
> > > Tighter relaxations: Relaxations that are tighter than Planet have been introduced (Tjandraatmadja et al., 2020, [B]), and are being used already (see [A]). And since [A] was accepted in ICLR'21, we cannot ignore it. The solver designed by the authors in this paper is customised for Planet relaxation and I cannot see a direct extension to the tighter relaxation.
> > > Further, [B] showed that propagation-based solvers can be designed for tighter relaxations too. There is no need for comparison with B, but a comparison with optimized-LiRPA is required.
> > > Thus, it becomes difficult to evaluate your work in regard to propagation-based solvers unless you compare with optimized LiRPA.
> > >
> > > [B]: Overcoming the Convex Barrier for Simplex Inputs, NeurIPS'21

---

### Official Review · Reviewer_ZCXL · 2021-11-05

**Correctness:** 3
**Technical Novelty And Significance:** 2
**Empirical Novelty And Significance:** 3
**Recommendation:** 5
**Confidence:** 3

**Main Review:**

While this paper is well-written and has proposed quite an interesting idea to improve the scalability of the neural network verification problem, the idea of applying operator or variable splitting to neural networks is definitely not novel, although neural network training and verification might be differet tasks. However, due to their highly similar problem formulations, I woule expect the authors to discuss the similarities and difference between the proposed method and such related papers, e.g., Taylor et al. (2016), Cui et al. (2019), Zeng et al. (2019, 2021). I would raise my score if the authors can explain the above in their response and include such details in the revision of the paper.

---
Cui, Y., He, Z., & Pang, J. S. (2020). MultiComposite nonconvex optimization for training deep neural networks. SIAM Journal on Optimization, 30(2), 1693--1723.

Zeng, J., Lau, T. T. K., Lin, S., & Yao, Y. (2019). Global convergence of block coordinate descent in deep learning. In ICML.

Zeng, J., Lin, S. B., Yao, Y., & Zhou, D. X. (2021). On ADMM in deep learning: Convergence and saturation-avoidance. Journal of Machine Learning Research, 22(199), 1--67.


**Summary Of The Paper:**

This paper proposes an operator splitting method which solves a convex relaxation of a learge-scale nonconvex optimization problem for analyzing the worst-case performance of deep neural networks against input perturbations. The proposed method is modular, scalable and parallelizable. Experimentally, the authors demonstrate that the proposed method has tighter bounds on the worst-case performance of large CNNs in image classification and RL settings.

**Summary Of The Review:**

This paper has proposed an interesting idea (i.e., operator splitting) to improve the scalability of neural network verification, but this idea is not novel for its applications to neural networks (e.g., neural network training). The paper will better position itself if more detailed comparisons are made with such methods in the existing literature.

---

> ### Author Response · Authors · 2021-11-23
> **Reply to Reviewer ZCXL**
>
> We thank the reviewer for pointing out papers on training NNs using operator splitting. In the papers mentioned by the reviewer, the training loss function is to be minimized subject to equality constraints defined by the NN equations. The decision variables are the NN parameters as well as NN “states” (the out of activation layers+input to NN). This is a con-convex problem and the goal of operator splitting there is to distribute/parallelize the computations for finding a stationary point. For this problem, various forms of ADMM can be used (e.g., multi-block ADMM) since convergence to a global solution is not concerned.
>
> In NN verification, a function of the NN output is to be minimized subject to NN equations and the input to be inside a set (NN params are of course fixed). For this non-convex problem, we must find either the global solution or a *guaranteed bound* on the objective function using convex relaxations of the non-convex problem, which becomes a fundamentally different optimization problem. One cannot directly apply an operator splitting on the original non-convex problem (as done in NN training) or even on its convex relaxation. For example, applying a direct three-block extension of ADMM is not possible (due to lack of convergence guarantees [A]) and we must be careful with splitting so that the resulting convex problem satisfies the assumptions of ADMM for convergence and be efficiently implementable: The objective must be fully separable across two sets of decision variables and each of the subproblems must be easy to solve. We remark that the proposed splitting is of little similarity with the existing operator splitting methods for NN training, and the ADMM cannot be applied here in a plug-and-play fashion. Overall, we believe that the proposed variable splitting is technically novel from this perspective. We will include the similarities and differences in more detail in the revised paper.
>
> [A]: Chen, Caihua, et al. "The direct extension of ADMM for multi-block convex minimization problems is not necessarily convergent." Mathematical Programming 155.1-2 (2016): 57-79.

---

### Decision · Program_Chairs · 2022-01-20

**Decision:**

Reject

**Comment:**

The authors propose a novel operator splitting method for solving convex relaxations of neural network verification problems, and develop and validate an optimized implementation of the same on large scale networks, focusing on the problem of verifying robustness to norm bounded adversarial perturbations.

The reviewers agree that the paper contains interesting ideas that are worthy of further development and that these ideas may prove useful eventually in pushing the envelope of what is possible in neural network verification. However, in its current form, the paper misses some key experimental evidence to rigorously evaluate the value of the contributions made:
1) Comparison against SOTA incomplete verifiers: The authors do not provide detailed and rigorous comparisons against well-known baselines (for example the incomplete verifiers from Fast-and-complete (Xu et al., 2021), Beta-CROWN(Wang et al. 2021))
2) Incorporating tighter relaxations: It would be valuable for the community to understand whether the proposed algorithm is compatible with tighter relaxations like those of (Tjandraatmadja et al., 2020). Even if they are not, it would be interesting to understand the comparison against standard solvers for these tighter relaxations compared against the advanced solver developed by the authors applied to the weaker relaxation.
3) Showing performance in the context of complete verification: While this is not a requirement, it would be great to see how the method performs in the conjunction with a branch and bound search, as this sometimes reveals surprising tradeoffs or weaknesses of incomplete verifiers (as observed in the results of Beta-CROWN(Wang et al. 2021)).

I encourage the authors to strengthen the paper adding these experiments and resubmit to a future venue.